# Strip1 regulates retinal ganglion cell survival by suppressing Jun-mediated apoptosis to promote retinal neural circuit formation

**Mai Ahmed, Yutaka Kojima, Ichiro Masai\***

Developmental Neurobiology Unit, Okinawa Institute of Science and Technology Graduate University, Okinawa, Japan

**Abstract** In the vertebrate retina, an interplay between retinal ganglion cells (RGCs), amacrine (AC), and bipolar (BP) cells establishes a synaptic layer called the inner plexiform layer (IPL). This circuit conveys signals from photoreceptors to visual centers in the brain. However, the molecular mechanisms involved in its development remain poorly understood. Striatin-interacting protein 1 (Strip1) is a core component of the striatin-interacting phosphatases and kinases (STRIPAK) complex, and it has shown emerging roles in embryonic morphogenesis. Here, we uncover the importance of Strip1 in inner retina development. Using zebrafish, we show that loss of Strip1 causes defects in IPL formation. In *strip1* mutants, RGCs undergo dramatic cell death shortly after birth. AC and BP cells subsequently invade the degenerating RGC layer, leading to a disorganized IPL. Mechanistically, zebrafish Strip1 interacts with its STRIPAK partner, Striatin 3 (Strn3), and both show overlapping functions in RGC survival. Furthermore, loss of Strip1 or Strn3 leads to activation of the proapoptotic marker, Jun, within RGCs, and Jun knockdown rescues RGC survival in *strip1* mutants. In addition to its function in RGC maintenance, Strip1 is required for RGC dendritic patterning, which likely contributes to proper IPL formation. Taken together, we propose that a series of Strip1-mediated regulatory events coordinates inner retinal circuit formation by maintaining RGCs during development, which ensures proper positioning and neurite patterning of inner retinal neurons.

**\*For correspondence:**
masai@oist.jp

**Competing interest:** The authors declare that no competing interests exist.

## Editor's evaluation

The results provide mechanistic insight into Strip1 and Striatin-interacting phosphatase and kinase (STRIPAK) complex function at the cellular and molecular level in the developing retina. They show that a primary function of Strip1 and the larger STRIPAK complex in retinal ganglion cells is to promote survival by suppressing Jun-mediated apoptosis. Reviewers were most interested to know whether Jun-mediated, pro-apoptotic signaling occurs due to connectivity defects or if it is connectivity-independent, and the authors have recognized the difficulty in addressing this point, and conclude that it is unlikely that failure of connectivity in the inner plexiform layer is the cause of retinal ganglion cell death.

## Introduction

The retina is a highly organized neural circuit that comprises six major classes of neurons, assembled into three cellular layers with two synaptic or plexiform layers between them. This beautiful layered architecture is commonly referred to as 'retinal lamination' (*Avanesov and Malicki, 2010*; *D'Orazi et al., 2014*; *Dowling, 1987*). Lamination is conserved among vertebrates and is critical for processing

**eLife digest** The back of the eye is lined with an intricate tissue known as the retina, which consists of carefully stacked neurons connecting to each other in well-defined 'synaptic' layers. Near the surface, photoreceptors cells detect changes in light levels, before passing this information through the inner plexiform layer to retinal ganglion cells (or RGCs) below. These neurons will then relay the visual signals to the brain. Despite the importance of this inner retinal circuit, little is known about how it is created as an organism develops.

As a response, Ahmed et al. sought to identify which genes are essential to establish the inner retinal circuit, and how their absence affects retinal structure. To do this, they introduced random errors in the genetic code of zebrafish and visualised the resulting retinal circuits in these fast-growing, translucent fish. Initial screening studies found fish with mutations in a gene encoding a protein called Strip1 had irregular layering of the inner retina.

Further imaging experiments to pinpoint the individual neurons affected showed that in zebrafish without Strip1, RGCs died in the first few days of development. Consequently, other neurons moved into the RGC layer to replace the lost cells, leading to layering defects. Ahmed et al. concluded that Strip1 promotes RGC survival and thereby coordinates proper positioning of neurons in the inner retina.

In summary, these findings help to understand how the inner retina is wired; they could also shed light on the way other layered structures are established in the nervous system. Moreover, this study paves the way for future research investigating Strip1 as a potential therapeutic target to slow down the death of RGCs in conditions such as glaucoma.

visual information (*Baden et al., 2020*; *Nassi and Callaway, 2009*). During development, neurogenesis, cell migration, and neurite patterning are spatially and temporally coordinated to form retinal lamination. Any defect in these events can disrupt retinal wiring and consequently compromise visual function (*Amini et al., 2017*). However, molecular mechanisms that govern retinal neural circuit formation are not fully understood.

The retinal neural circuit processes visual signals through two synaptic neuropils (*Figure 1A*). At the apical side, the outer plexiform layer (OPL) harbors synapses that transmit input from photoreceptors (PRs) in the outer nuclear layer (ONL) to bipolar (BP) and horizontal cells (HCs) in the inner nuclear layer (INL). At the basal side of the retina, the inner plexiform layer (IPL) is densely packed with synaptic connections formed between BPs and amacrine cells (ACs) in the INL, and retinal ganglion cells (RGCs) in the ganglion cell layer (GCL). The retina contains one type of glial cells called Müller glia (MGs), which span the apicobasal axis of the retina (*Hoon et al., 2014*; *Huberman et al., 2010*).

RGCs are the first-born retinal neurons, which extend their axons to exit the eye cup and innervate visual centers in the brain (*D'Souza and Lang, 2020*, *Robles et al., 2014*). In mouse and zebrafish models, when RGCs are absent or exhibit defects in axon projections, vision is compromised (*Kay et al., 2001*; *Moshiri et al., 2008*; *Rick et al., 2000*). Therefore, RGCs are indispensable for vision. RGC degeneration is often a secondary defect in optic neuropathies and one of the leading causes of blindness worldwide. Thus, tremendous efforts are being dedicated to deciphering signaling pathways involved in RGC death (*Almasieh et al., 2012*; *Maes, 2017*; *Munemasa and Kitaoka, 2012*).

Striatin interacting protein 1 (Strip1) is a recently identified protein with emerging functions in neuronal development. It was first described as one of the core components of the striatin-interacting phosphatases and kinases (STRIPAK) complex (*Goudreault et al., 2009*). The STRIPAK complex is an evolutionarily conserved supramolecular complex with diverse functions in cell proliferation, migration, vesicular transport, cardiac development, and cancer progression (*He et al., 2010*; *Hwang and Pallas, 2014*; *La marca, 2019*; *Madsen et al., 2015*; *Neisch et al., 2017*; *Shi et al., 2016*). In addition, several STRIPAK components participate in dendritic development, axonal transport, and synapse assembly (*Chen et al., 2012*; *Li et al., 2018*; *Schulte et al., 2010*). In *Drosophila*, Strip (a homolog of mammalian Strip1/2) is essential for axon elongation by regulating early endosome trafficking and microtubule stabilization (*Sakuma et al., 2014*; *Sakuma et al., 2015*). In addition, Strip, together with other STRIPAK members, modulates synaptic bouton development and prevents ectopic retina formation (*Neal et al., 2020*; *Sakuma et al., 2016*). On the other hand, loss of mouse Strip1 causes

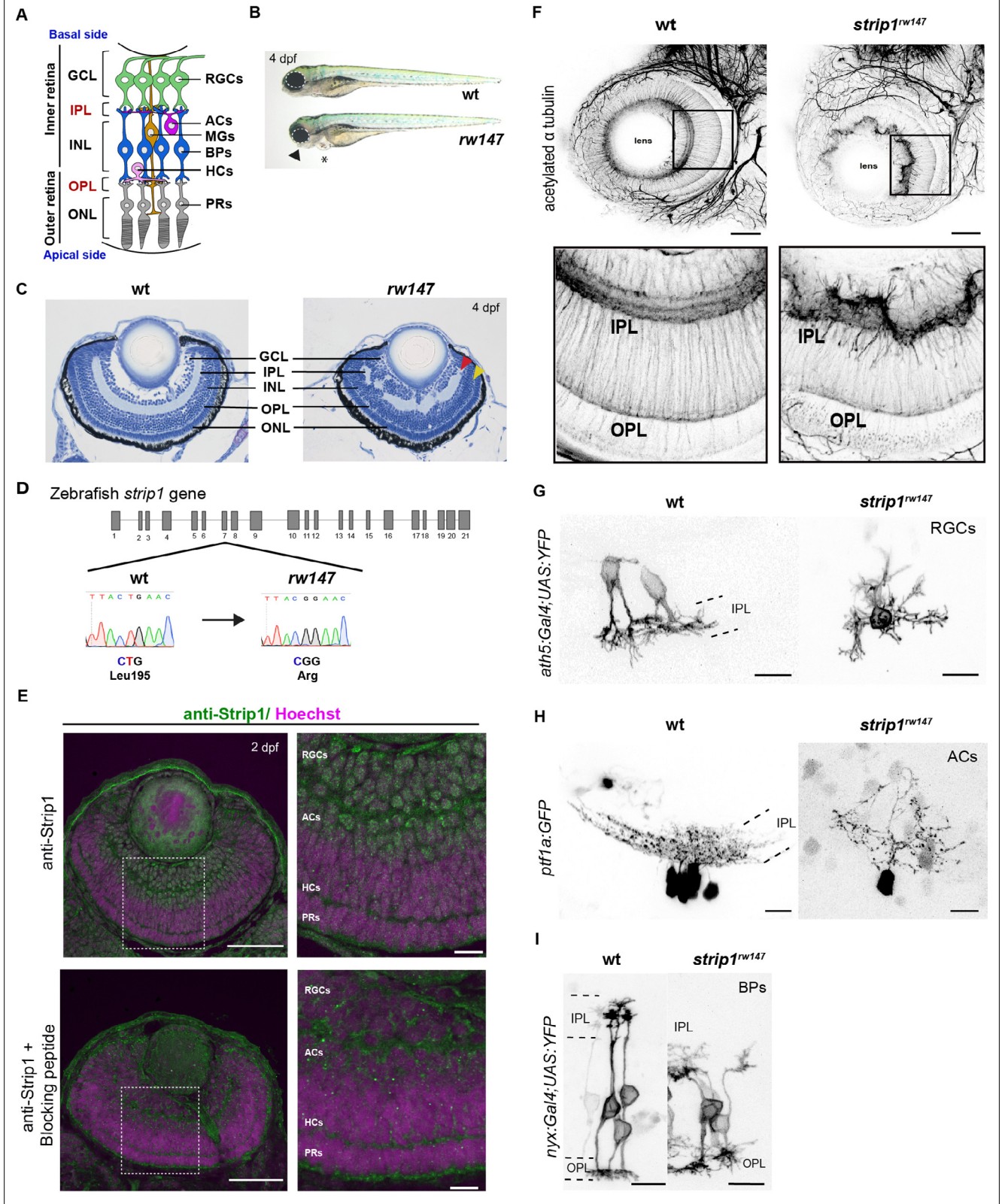

**Figure 1.** Striatin-interacting protein 1 (Strip1) is essential for inner retinal neural circuit development. (**A**) Zebrafish retinal neural circuit showing retinal neurons and synaptic layers. GCL, ganglion cell layer; IPL, inner plexiform layer; INL, inner nuclear layer; OPL, outer plexiform layer; ONL, outer nuclear layer; RGCs, retinal ganglion cells; ACs, amacrine cells; BPs, bipolar cells; HCs, horizontal cells; PRs, photoreceptors; MGs, Müller glia. (**B**) Morphology of wild-type and *rw147* embryos at 4 dpf. Dotted lines demarcate the eye. An arrowhead indicates abnormal lower jaw. An asterisk indicates heart

*Figure 1 continued on next page*

*Figure 1 continued*

edema. (**C**) Wild-type and *rw147* mutant retinas at 4 dpf. Red and yellow arrowheads indicate the IPL and OPL, respectively. (**D**) A missense mutation occurs in *strip1* gene of *rw147* mutants leading to replacement of Leu195 with arginine. (**E**) Wild-type retinas labeled with anti-Strip1 antibody (upper panels) and anti-Strip1 plus Strip1-blocking peptide as a negative control (lower panels). Nuclei are stained with Hoechst. Scale bar, 50 μm. Right panels show higher magnification of outlined areas. Scale bar, 10 μm. (**F**) Whole-mount labeling of 3-dpf wild-type and *strip1^rw147* mutant retinas with anti-acetylated α-tubulin antibody. Bottom panels show higher magnification of outlined areas. Scale bar, 50 μm. (**G**) Projection images of single RGCs at 2 dpf expressing *ath5:Gal4VP16; UAS:MYFP* in wild-type and *strip1^rw147* mutants. Scale bar, 10 μm. (**H**) Projection images of single ACs at 3 dpf expressing *ptf1a:GFP* in wild-type and *strip1^rw147* mutants. Scale bar, 10 μm. (**I**) Projection images of single BPs at 3 dpf expressing *nyx:Gal4VP16; UAS:MYFP* in wild-type and *strip1^rw147* mutants. Scale bar, 10 μm.

The online version of this article includes the following source data and figure supplement(s) for figure 1:

**Figure supplement 1.** Retinal lamination defects in *strip1^rw147* mutants are phenocopied by either CRISPR- or MO-mediated knockdown of *strip1* and rescued by overexpression of Strip1.

**Figure supplement 1—source data 1.** Data for *Figure 1—figure supplement 1D*.

**Figure supplement 1—source data 2.** Data for *Figure 1—figure supplement 1G*.

**Figure supplement 1—source data 3.** Data for *Figure 1—figure supplement 1I*.

**Figure supplement 2.** *strip1* mRNA is expressed in developing retinas and is required for inner plexiform layer (IPL) formation.

early mesoderm migration defects leading to embryonic lethality (*Bazzi et al., 2017*; *Zhang et al., 2021*). Thus, the role of Strip1 in the vertebrate nervous system is largely unknown.

Here, we report an essential role for Strip1 in neural circuit formation of zebrafish retina. In zebrafish *strip1* mutants, retinal lamination, especially IPL formation, is disrupted. Loss of Strip1 causes RGC death shortly after birth. Cells in the INL subsequently infiltrate the degenerating GCL, leading to a disorganized IPL. Strip1 cell autonomously promotes RGC survival; however, it is not required in INL cells for IPL formation. Therefore, Strip1-mediated RGC maintenance is required to establish the IPL. Mechanistically, we identified Striatin 3 (Strn3) as a Strip1-interacting partner. Both Strip1 and Strn3 show overlapping functions in RGC survival through suppression of the Jun-mediated apoptotic pathway. We also found that Strip1 is cell autonomously required for RGC dendritic patterning, which likely promotes interaction between RGCs and ACs for IPL formation. Collectively, we demonstrate that Strip1 is crucial for RGC survival during development and thereby coordinates proper wiring of the inner retina.

## Results

### Strip1 is essential for inner retinal neural circuit development

To understand mechanisms of retinal neural circuit formation, we screened zebrafish retinal lamination-defective mutants (*Masai et al., 2003*) and identified the *rw147* mutant. At 4 days post-fertilization (dpf), *rw147* mutant embryos have small eyes, lower jaw atrophy, and cardiac edema (*Figure 1B*). *rw147* mutants also show defects in retinal lamination, in which retinal layers, especially in the inner retina, fluctuate in a wave-like pattern (*Figure 1C*). The *rw147* mutation is lethal by 6 dpf due to cardiac edema. Mapping of the *rw147* mutation revealed a missense mutation in exon 7 of the *strip1* gene of the *rw147* mutant genome (*Figure 1D*).

Next, we performed CRISPR-Cas9-mediated mutagenesis to generate a 10-base deletion mutant, *strip1^crisprΔ10* (*Figure 1—figure supplement 1A*). *strip1^crisprΔ10* mutants show similar morphology and retinal lamination defects to those of *strip1^rw147* mutants (*Figure 1—figure supplement 1B–D*). Likewise, knockdown of Strip1 using translation-blocking morpholinos (MO-strip1) phenocopied *strip1^rw147* mutants (*Figure 1—figure supplement 1E, F*). We verified the specificity of MO-strip1 using a custom-made zebrafish Strip1 antibody that fails to detect a 93 kDa protein band corresponding to zebrafish Strip1 in the morphants (*Figure 1—figure supplement 1G*). Furthermore, we generated transgenic lines that express wild-type and *rw147* mutant forms of zebrafish Strip1 protein under the control of the heat shock promotor, *Tg[hsp:WT.Strip1-GFP]* and *Tg[hsp:Mut.Strip1-GFP]*, respectively. Wild-type Strip1, but not the mutant form, rescued the retinal defects of *strip1^rw147* (*Figure 1—figure supplement 1H, I*). Taken together, the *strip1* mutation is the cause of retinal lamination defects.

Next, we examined Strip1 expression in wild-type retinas by labeling with the zebrafish Strip1 antibody. Strip1 was expressed in RGCs and ACs at 2 dpf (*Figure 1E*). In situ hybridization shows that

*strip1* mRNA is maternally and zygotically expressed and by 2 dpf, expression becomes restricted to the eyes, optic tectum, and heart (*Figure 1—figure supplement 2A, B*). Like Strip1 protein, *strip1* mRNA was expressed in RGCs and ACs (*Figure 1—figure supplement 2C*). To visualize retinal neuropils, we performed whole-mount staining of the retina with anti-acetylated α-tubulin antibody. In wild-type retinas, IPL and OPL are evident at 3 dpf. In contrast, IPL shows abnormal morphology, whereas OPL is relatively normal in *strip1$^{rw147}$* mutants (*Figure 1F*). We tracked IPL development using Bodipy TR stain. In wild-type retinas, a rudimentary IPL was formed as early as 52 hr post-fertilization (hpf); however, it was less defined in *strip1$^{rw147}$* mutants (*Figure 1—figure supplement 2D*). At 62 hpf, mutants exhibited a wave-like IPL. This temporal profile coincides with development of RGCs and ACs. Next, we visualized neurite morphology of RGCs, ACs, and BPs by transiently expressing fluorescent proteins under control of *ath5* (*Masai et al., 2003*), *ptf1a* (*Jusuf and Harris, 2009*), and *nyx* promoters (*Schroeter et al., 2006*), respectively. In wild-type siblings, RGCs and ACs normally extend their dendrites toward the IPL; however, *strip1$^{rw147}$* mutants show randomly directed dendritic patterns of RGCs and ACs (*Figure 1G, H*). In wild-type siblings, BPs normally extend their axons and dendrites toward IPL and OPL, respectively; however, BPs of *strip1$^{rw147}$* mutants show misrouted axons and abnormal dendritic branching (*Figure 1I*). Thus, Strip1 is required for IPL formation and correct neurite patterning of RGCs, ACs, and BPs.

## RGCs are reduced and INL cells infiltrate the GCL in *strip1* mutants

To examine how the IPL is disrupted in *strip1* mutants, we combined *strip1$^{rw147}$* mutants with two transgenic lines, *Tg[ath5:GFP; ptf1a:mCherry-CAAX]*, to visualize RGCs and ACs. In *Tg[ath5:GFP]*, GFP is expressed strongly in RGCs and weakly in ACs and PRs under control of the *ath5* enhancer (*Masai et al., 2003*). In *Tg[ptf1a:mCherry-CAAX]*, membrane-targeted mCherry is expressed in ACs and HCs under control of *ptf1a* promoter (*Jusuf and Harris, 2009*). Live imaging of 3-dpf retinas revealed that RGCs are severely reduced in *strip1$^{rw147}$* mutants (*Figure 2A*). Since we observe a slight reduction in total retinal area of *strip1$^{rw147}$* mutants at 3 dpf (*Figure 2—figure supplement 1A, B*), we quantified RGC area compared to total retinal area and found that mutant ath5+ RGCs are reduced, reaching only 7.45% ± 2.88% of total retinal area, compared to 24.18% ± 1.48% in wild-type siblings (*Figure 2B*). On the other hand, there was no significant change in the number of ptf1a+ ACs between *strip1$^{rw147}$* mutants and wild-type siblings (*Figure 2A, C*). However, ptf1a+ ACs abnormally extended their dendrites to form an irregularly patterned IPL (*Figure 2A*, asterisks). In wild-type retinas, the majority of ACs reside in the INL, except displaced ACs (*Jusuf and Harris, 2009*). However, in *strip1$^{rw147}$* mutants, a significant fraction of ptf1a+ ACs were abnormally located in the GCL (*Figure 2A*, arrowheads in bottom panels, and *Figure 2D*). Such abnormal positioning of ACs is correlated with the severity of IPL defects (*Figure 2—figure supplement 1C*). This phenotype is reminiscent of the *ath5* mutant, *lakritz*, in which RGCs fail to undergo neurogenesis, leading to infiltration of ACs into the GCL and transient IPL formation defects (*Kay et al., 2001*; *Kay et al., 2004*). We confirmed similar IPL defects in *ath5* morphant retinas at 3 dpf (*Figure 2—figure supplement 1D*), albeit weaker than those of *strip1$^{rw147}$* mutants.

Next, we visualized ACs using anti-Pax6 antibody, which strongly labels ACs and weakly labels RGCs (*Macdonald and Wilson, 1997*). In wild-type siblings, most strong Pax6+ cells were in the INL, and only 9.84% ± 4.13% were in the GCL (*Figure 2E, G*). However, in *strip1$^{rw147}$* mutants, a significant percentage of Pax6+ cells (44.26% ± 17.8%) was in the GCL (*Figure 2E, G*). The total number of Pax6+ cells did not differ between wild-type siblings and *strip1$^{rw147}$* mutants (*Figure 2F*). We confirmed the abnormal positioning of ACs in the GCL using anti-parvalbumin, which labels subsets of ACs in the INL, together with displaced ACs in the GCL (*Maurer et al., 2010*; *Figure 2—figure supplement 1E-G*). Next, we visualized BPs using anti-Prox1 antibody, which labels BPs and HCs (*Jusuf and Harris, 2009*). In wild-type, 100% of Prox1+ cells were in the INL (*Figure 2H, J*). However, 10.6% ± 6.26% of Prox1+ cells were abnormally located in the GCL in *strip1$^{rw147}$* mutants (*Figure 2H, J*). The total number of Prox1+ cells did not differ between wild-type siblings and *strip1$^{rw147}$* mutants (*Figure 2I*). We performed labeling of double-cone and rod PRs using zpr1 and zpr3 antibodies, respectively (*Nishiwaki et al., 2008*). Apart from occasional mildly disrupted areas, the PR cell layer was grossly intact, with no positioning defects (*Figure 2—figure supplement 2A, B*). MG and proliferating cells at the ciliary marginal zone were visualized using anti-glutamine synthetase (GS) (*Peterson et al., 2001*) and anti-PCNA antibodies (*Raymond et al., 2006*), respectively. Both cell types showed grossly

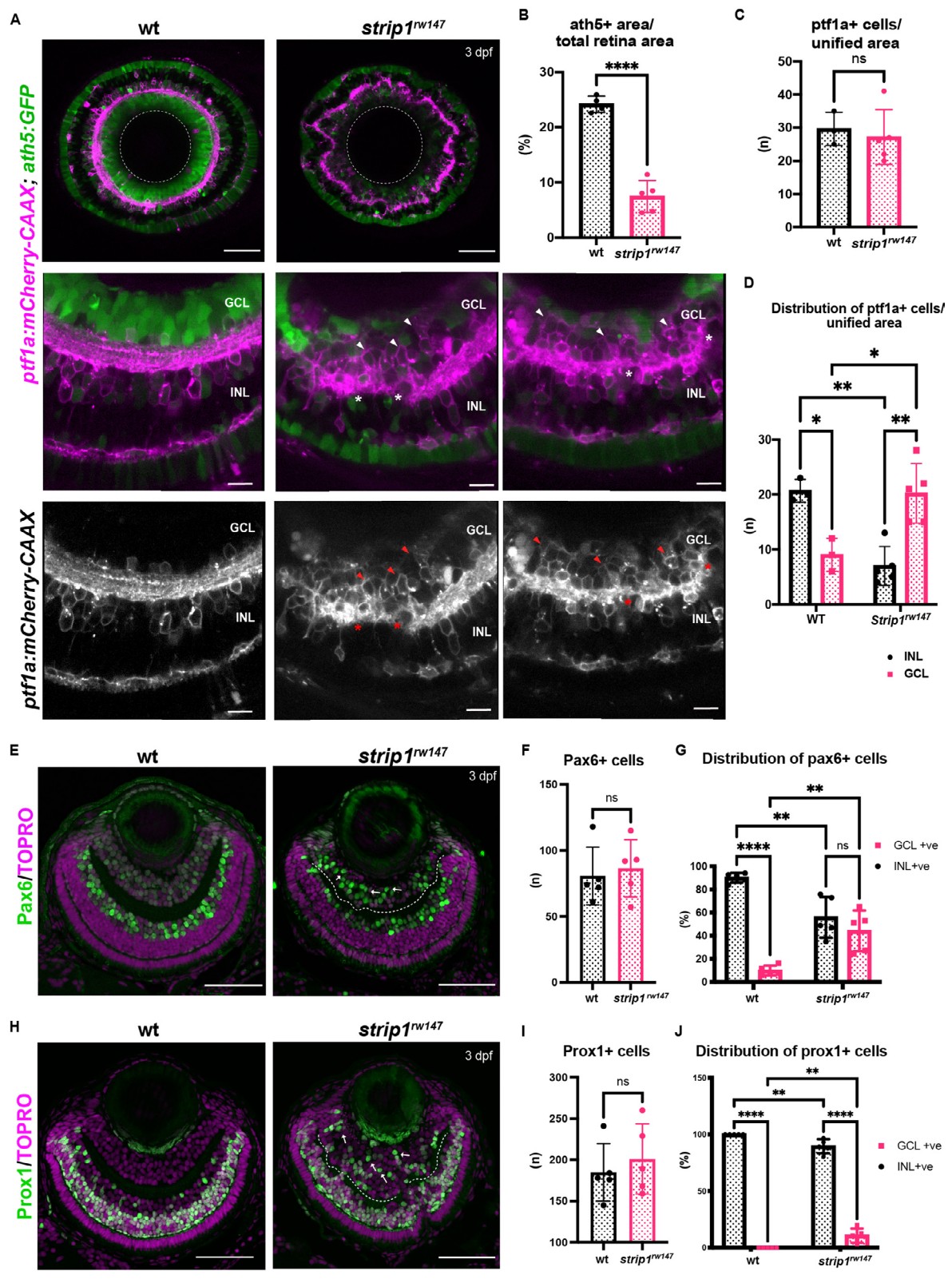

**Figure 2.** Retinal ganglion cells (RGCs) are reduced and INL cells infiltrate the GCL in *strip1* mutants. (**A**) Confocal sections of wild-type and *strip1^rw147* mutant retinas combined with the transgenic line *Tg[ath5:GFP; ptf1a:mCherry-CAAX]* to label RGCs and amacrine cells (ACs). Middle panels represent higher magnification. Lower panels show the magenta channel. Arrowheads indicate abnormal positioning of ptf1a+ ACs in the GCL. Asterisks show AC dendritic patterning defects. INL, inner nuclear layer; GCL, retinal ganglion cell layer. Scale bars, 50 µm (upper panels) and 10 µm (middle and lower

*Figure 2 continued on next page*

*Figure 2 continued*

panels). (**B**) Percentage of ath5+ area relative to total retinal area. Student's *t*-test with Welch's correction, $n \geq 4$. (**C**) AC numbers per unified retinal area (8500 µm²). Student's *t*-test with Welch's correction, $n \geq 3$. (**D**) Distribution of ACs (GCL or INL) per unified retinal area (8500 µm²). Two-way analysis of variance (ANOVA) with the Tukey multiple comparison test, $n \geq 3$. (**E**) Wild-type and *strip1^rw147* mutant retinas at 3 dpf labeled with anti-Pax6 antibody which strongly labels ACs. Arrows indicate strong Pax6+ cells that infiltrate the GCL. Nuclei are stained with TOPRO3. Scale bar, 50 µm. (**F**) The number of strong Pax6+ cells per retina. Student's *t*-test with Welch's correction, $n = 5$. (**G**) Percentage of strong Pax6+ cells (GCL+ or INL+) to the total number of strong Pax6+ cells. Two-way ANOVA with the Tukey multiple comparison test, $n = 5$. (**H**) Wild-type and *strip1^rw147* mutant retinas at 3 dpf labeled with anti-Prox1 antibody. Arrows indicate Prox1+ cells that infiltrate the GCL. Nuclei are stained with TOPRO3. Scale bar, 50 µm. (**I**) The number of Prox1+ cells per retina. Student's *t*-test with Welch's correction, $n = 5$. (**J**) Percentage of Prox1+ cells (GCL+ or INL+) to the total number of Prox1+ cells. Two-way ANOVA with the Tukey multiple comparison test, $n = 5$. For all graphs, data are represented as means ± standard deviation (SD). ns, not significant, *p < 0.05, **p < 0.01, and ****p < 0.0001.

The online version of this article includes the following source data and figure supplement(s) for figure 2:

**Source data 1.** Data for *Figure 2BCDFGIJ*.

**Figure supplement 1.** Amacrine cells (ACs) are abnormally positioned at the basal side of the inner plexiform layer (IPL) in *strip1* mutants and *ath5* morphants.

**Figure supplement 1—source data 1.** Data for *Figure 2—figure supplement 1B,F,G*.

**Figure supplement 2.** Photoreceptors, Müller glia, and ciliary marginal zone (CMZ) are not grossly affected by *strip1* mutation.

normal positioning in *strip1^rw147* mutants (*Figure 2—figure supplement 2C, D*). Thus, in the absence of Strip1, INL cells abnormally infiltrate the GCL and seem to replace the reduced RGCs.

## Strip1 cell autonomously promotes RGC survival

In zebrafish, RGC genesis starts in the ventronasal retina at 25 hpf, spreads into the entire retina by 36 hpf and is completed by 48 hpf (*Avanesov and Malicki, 2010*; *Hu and Easter, 1999*). Reduction of RGCs in *strip1* mutants could be due to compromised RGC genesis or RGC death after birth. To clarify which, we examined RGC genesis by monitoring ath5:GFP expression, and apoptosis by terminal deoxynucleotidyl transferase dUTP nick end labeling (TUNEL). In *strip1^rw147* mutants, RGCs are normally produced at 36 hpf; however, apoptosis occurred in the GCL at 48 hpf (*Figure 3A*). The number of apoptotic cells in GCL reached its highest level at 60 hpf, and apoptotic cells were eliminated by 96 hpf (*Figure 3A, B*). Accordingly, RGC population was significantly lower in *strip1^rw147* mutants than in wild-type siblings at 60 hpf and progressively reduced by 96 hpf (*Figure 3C*). In contrast, other retinal layers of *strip1^rw147* mutants showed slightly, but not significantly increased apoptosis at 72 hpf (*Figure 3—figure supplement 1A*), suggesting a specific function of Strip1 in RGC survival. In addition, despite the reduction in ath5:GFP+ area, the total presumptive GCL area, which was defined by retinal area between the lens and the IPL, was unchanged in *strip1^rw147* mutants throughout the stages (*Figure 3—figure supplement 1B*), suggesting that infiltrating INL cells replace the lost RGCs. We confirmed RGC death in *strip1^crispr∆10* mutants (*Figure 3—figure supplement 1C, D*). Interestingly, we observed apoptosis in the optic tectum of *strip1^rw147* mutants (*Figure 3—figure supplement 1E, F*), suggesting a common Strip1-dependent survival mechanism in the optic tectum. RGCs are the only retinal neurons which project their axons to the optic tectum. In *strip1^rw147* mutants, RGC axons appeared to exit from the eye cup and formed an optic chiasm at 3 dpf (*Figure 3—figure supplement 1G*). However, consistent with the reduction of RGCs, the optic nerve was thinner in *strip1^rw147* mutants than in wild-type siblings and showed elongation defects toward the optic tectum.

To determine whether Strip1 cell autonomously promotes RGC survival, we conducted cell transplantation from *strip1^rw147* mutant donor cells into wild-type host embryos at the blastula stage. TUNEL of transplanted retinas at 60 hpf revealed that *strip1^rw147* mutant donor RGCs underwent apoptosis in wild-type host retinas (*Figure 3D–F*). To address whether Strip1 is also required for RGC neurite development, we repeated the same experiment using mutant donors carrying the transgene *ath5:GFP*, to examine RGC neurite patterns (*Figure 3—figure supplement 2A*). Visualization was performed at 57–58 hpf, when wild-type RGCs exhibit apically projected dendrites, while mutant RGCs had not yet undergone complete degeneration. As expected, wild-type transplanted RGCs display uniform dendritic patterns projecting toward the nascent IPL (*Figure 3—figure supplement 2B*). We can also observe several RGCs projecting their axons basally (arrowheads, *Figure 3—figure supplement 2B*). However, the majority of mutant RGCs transplanted in wild-type retina show irregular neurite projections, apically directed processes (presumably dendrites) do not project to a uniform layer and show

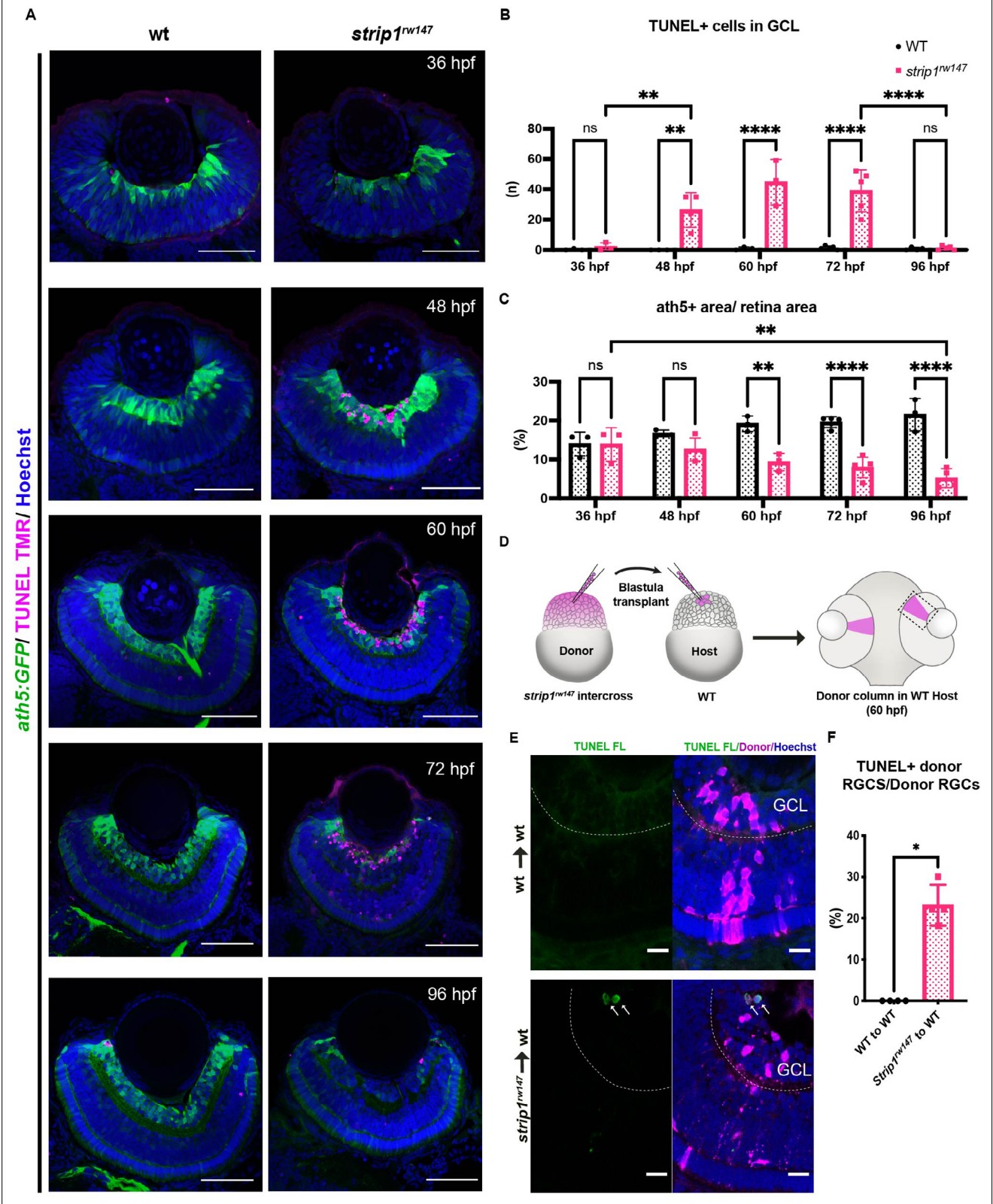

**Figure 3.** Strip1 cell autonomously promotes retinal ganglion cell (RGC) survival. (**A**) Transferase dUTP nick end labeling (TUNEL) of wild-type and *strip1*^rw147 mutant retinas carrying the transgene *Tg[ath5:GFP]* to label RGCs. Nuclei are stained with Hoechst. Scale bar, 50 µm. (**B**) The number of TUNEL+ cells in ganglion cell layer (GCL). Two-way analysis of variance (ANOVA) with the Tukey multiple comparison test, *n* ≥ 3. (**C**) Percentage of ath5+ area relative to total retinal area. Two-way ANOVA with the Tukey multiple comparison test, *n* ≥ 3. (**D**) Cell transplantation design to evaluate the cell

*Figure 3 continued on next page*

*Figure 3 continued*

autonomy of Strip1 in RGC survival. Donor embryos from a *strip1^{rw147}* mutant background are labeled with dextran rhodamine and transplanted into host wild-type embryos. Hosts that show transplanted retinal columns at 60 hpf were subjected to TUNEL. (**E**) 60-hpf host retinas stained with TUNEL FL to visualize apoptotic cells in wild type to wild type (upper panel) or *strip1^{rw147}* mutant to wild type (lower panel). Arrows indicate the presence of apoptotic donor cells. Scale bar, 10 μm. (**F**) Percentage of TUNEL+ donor RGCs relative to total donor RGCs. Mann–Whitney *U*-test, *n* = 4. For all graphs, data are represented as means ± SD. *p < 0.05, **p < 0.01, and ****p < 0.0001.

The online version of this article includes the following source data and figure supplement(s) for figure 3:

**Source data 1.** Data for *Figure 3B,C,F*.

**Figure supplement 1.** *strip1* mutants show apoptosis in retinal ganglion cells (RGCs) and optic tectum, and elongation defects in retinal axons.

**Figure supplement 1—source data 1.** Data for *Figure 3—figure supplement 1A,B,D,F*.

**Figure supplement 2.** Strip1 is cell autonomously required to promote retinal ganglion cell (RGC) dendritic patterning.

distant abnormal branching (asterisks, *Figure 3—figure supplement 2C*). We also observe defects in basally directed neurites (probably axons), like bifurcation and misrouting (arrowheads, *Figure 3—figure supplement 2C*). Taken together, Strip1 is cell autonomously required for survival and neurite morphogenesis of RGCs.

## RGC death triggers abnormal positioning of ACs, leading to IPL disruption

ACs are proposed to be the main cell type responsible for IPL formation (*Godinho et al., 2005*; *Huberman et al., 2010*). To clarify how RGC death influences infiltration of ACs into GCL and IPL disruption, we performed time-lapse imaging of wild-type and *strip1^{rw147}* mutant retinas combined with the transgenic line *Tg[ath5:GFP; ptf1a:mCherry-CAAX]*. At 48 hpf, there were no apparent differences in position or morphology of RGCs and ACs between wild-type siblings and *strip1^{rw147}* mutants (*Figure 4A* and *Figure 4—videos 1; 2*). In *strip1^{rw147}* mutants at 52 hpf, RGCs started to disappear, creating an empty spot in the GCL (*Figure 4A*, asterisks). However, ACs were still located in the INL. At 55 hpf, a rudimentary IPL was observed in the central retina of both wild-type siblings and *strip1^{rw147}* mutants. At 59 hpf, ACs started to invade the empty spaces in the GCL (*Figure 4A*, arrowheads). Infiltration of ACs into the GCL was more prominent at 62 hpf, resulting in a fluctuating IPL. Thus, loss of RGCs triggers infiltration of ACs into the GCL in *strip1^{rw147}* mutants.

To examine whether Strip1 is required in ACs for IPL formation, we performed cell transplantation using donor embryos carrying the transgene *Tg[ptf1a:mCherry-CAAX]* (*Figure 4B*). When mutant ACs were transplanted into wild-type host retinas, most donor ACs were normally positioned in the INL and extended dendrites toward the IPL, as in the case of wild-type donor ACs transplanted into a wild-type host retina (*Figure 4C, D*). Occasionally, three ACs extended two dendritic trees instead of 1 among 73 transplanted ACs; however, such dendritic misprojection did not perturb IPL formation (*Figure 4—figure supplement 1A*). On the other hand, as with mutant donor ACs transplanted into mutant host retinas, when wild-type donor ACs were transplanted to mutant host retinas, they showed irregular neurite projection with many somas abnormally located toward the basal side, resulting in IPL formation defects (*Figure 4C, D* and *Figure 4—figure supplement 1A*). These data suggest a non-cell autonomous function of Strip1 in ACs for IPL formation.

Similarly, we conducted cell transplantation to assess the role of Strip1 in BPs. Mutant donor BPs labeled with the transgene *Tg[xfz43]* (*Zhao et al., 2009*) projected axons normally toward the IPL in wild-type host retinas, in the same fashion as wild-type donor BPs (*Figure 4—figure supplement 1B–D*). Few transplanted columns of mutant donors showed extra lateral branching and excessive elongation of BP arbors (*Figure 4—figure supplement 1D*, arrows). However, such arbor defects did not disrupt the IPL. On the other hand, when wild-type donor BPs labeled with the transgene, *Tg[xfz3]*, were transplanted into a mutant host retina, wild-type donor BP axons failed to project toward the mutant host IPL, but rather seemed to be guided toward the wild-type donor IPL (*Figure 4—figure supplement 1E–G*). Thus, Strip1 is not required in ACs or BPs for neurite projection to the IPL, although we do not exclude the possibility that Strip1 is cell autonomously required in a small subset of ACs and BPs to regulate dendritic branching and neurite extension. Taken together, it is likely that Strip1-mediated RGC maintenance is essential for proper neurite patterning of ACs and BPs, and for subsequent IPL formation.

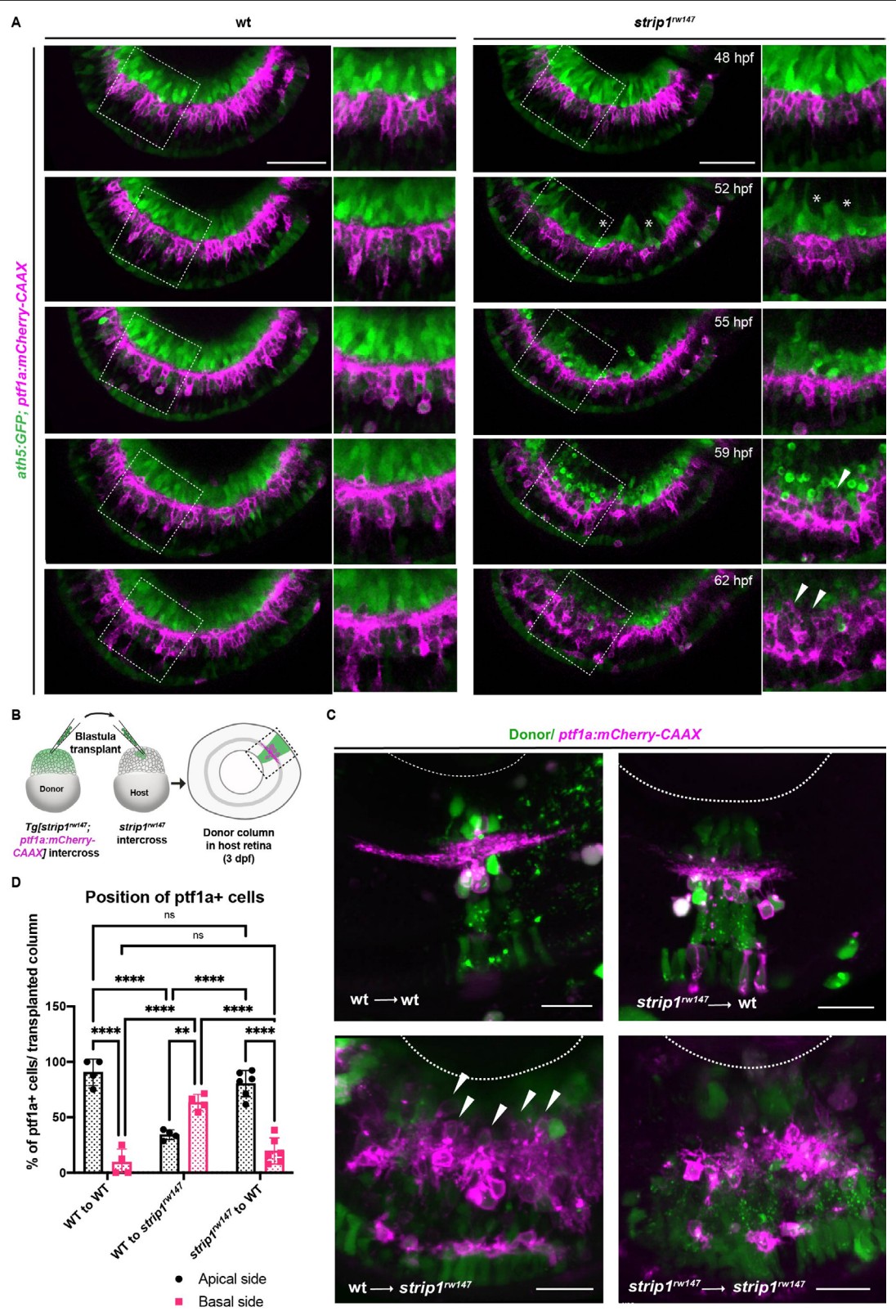

**Figure 4.** Retinal ganglion cell (RGC) death triggers abnormal positioning of amacrine cells (ACs) leading to inner plexiform layer (IPL) disruption. (**A**) Time-lapse imaging of wild-type and *strip1^rw147* mutant retinas combined with the transgenic line *Tg[ath5:GFP; ptf1a:mCherry-CAAX]* to track ACs and RGCs during IPL formation. Asterisks denote empty areas in the ganglion cell layer (GCL). Arrowheads represent infiltration of ACs into empty spaces in the GCL. Panels on the right show higher magnification of outlined areas. Scale bar, 50 μm. (**B**) Cell transplantation design to evaluate the cell autonomy

*Figure 4 continued on next page*

*Figure 4 continued*

of Strip1 in AC-mediated IPL formation. Donor embryos are from intercross of *strip1^rw147* heterozygous fish combined with *Tg[ptf1a:mCherry-CAAX]* to label ACs. Host embryos are generated by nontransgenic intercross of *strip1^rw147* heterozygous fish. Donor cells are labeled with dextran Alexa-488 and transplanted into host embryos to make chimeric host retinas with donor-derived retinal columns. (C) Confocal images of four combinations of transplantation outcomes: wild type to wild type, wild type to mutant, mutant to wild type, and mutant to mutant. Arrowheads indicate abnormal positioning of ACs in basal side of IPL. Scale bar, 20 μm. (D) Percentage of ACs (either at the apical or the basal side of the IPL) relative to the total number of ACs within a transplanted column. Two-way analysis of variance (ANOVA) with the Tukey multiple comparison test, $n \geq 4$. Data are represented as means ± standard deviation (SD). $**p < 0.01$ and $****p < 0.0001$.

The online version of this article includes the following video, source data, and figure supplement(s) for figure 4:

**Source data 1.** Data for *Figure 4D*.

**Figure supplement 1.** Strip1 is not required in amacrine cells (ACs) and bipolar cells (BPs) for their neurite projections to the inner plexiform layer (IPL).

**Figure 4—video 1.** Development of amacrine cells (ACs) and inner plexiform layer (IPL) formation in wild-type sibling retina.
https://elifesciences.org/articles/74650/figures#fig4video1

**Figure 4—video 2.** Development of amacrine cells (ACs) and inner plexiform layer (IPL) formation in *strip1^rw147* mutant retina.
https://elifesciences.org/articles/74650/figures#fig4video2

## Strn3 is a Strip1-interacting partner that promotes RGC survival

To identify which molecules interact with Strip1 to regulate RGC survival, we conducted a co-immunoprecipitation experiment coupled with mass spectrometry (Co-IP/MS). Head lysates of wild-type embryos combined with the transgenic line *Tg[hsp:WT.Strip1-GFP]* were used to pull-down wild-type Strip1 and its interacting partners. As a negative control, we used lysates from two other lines: *Tg[hsp:Mut.Strip1-GFP]* and *Tg[hsp:Gal4;UAS:GFP]*, to rule out proteins enriched by the mutant form of Strip1 or by GFP alone (*Figure 5A, B*). Six proteins were enriched only by the wild-type form of Strip1, 5 of which are components of the STRIPAK complex (*Figure 5C, D*, and *Figure 5—figure supplement 1A, B*). Since none of these components has been studied in zebrafish, we analyzed previously published single-cell RNA sequencing data on transcriptomes from zebrafish embryonic retinas at 2 dpf (*Xu et al., 2020*), and found that only *strip1* and *strn3* mRNA are abundantly expressed in retinal cells (*Figure 5—figure supplement 1C–I*). Next, we knocked down Strn3 using a translation-blocking morpholino (MO-strn3), and we confirmed the specificity of knock down using a commercial anti-Strn3 antibody that shows a significant reduction of a ~90 kDa band corresponding to Strn3 in 2-dpf morphants (*Figure 5—figure supplement 2A, B*). We observed a significant increase in apoptotic cells in the GCL of *strn3* morphants at 60 hpf (*Figure 5E, F*). This leads to a significant reduction in RGCs at 60 and 76 hpf, as assessed by the *ath5:GFP* signal (*Figure 5G–I*). Although *strn3* morphants showed a similar RGC loss to *strip1* mutants, it was weaker. IPL defects were also milder in *strn3* morphants than in *strip1^rw147* mutants at 76 hpf (*Figure 5H*). Such observed weak phenotypes suggest that Strn3 may function in discrete RGCs populations or they could be due to diluted effects of MO-strn3 by 3 dpf. Taken together, Strn3 is a Strip1-interacting partner that shows similar roles in promoting RGC survival.

## Jun is a key mediator of RGC death in the absence of Strip1

To determine what kinds of molecules mediate RGC apoptosis in *strip1* mutants, we performed RNA sequencing on transcriptomes from 62-hpf eye cups of *strip1^rw147* mutants. Compared to wild-type siblings, *strip1* mutants had 131 significantly upregulated genes and 75 downregulated genes (*Figure 6A*, $\log_2 FC > |1|$, and False Discovery Rate (FDR) < 0.05). Most downregulated genes were markers of RGCs, like *isl2b*, *pou4f3* (also known as *brn3c*), and *tbr1b*, which reflects the reduction in RGCs. Genes related to synaptic development and transmission were also downregulated (*Figure 6A, B* and *Figure 6—figure supplement 1A*). On the other hand, many significantly upregulated genes were related to apoptosis, oxidative phosphorylation, cellular response to stress, and the MAP kinase (MAPK) signaling pathway (*Figure 6A, B* and *Figure 6—figure supplement 1B*). Most reports of RGCs undergoing stress are in glaucoma and optic nerve injury (ONI) models, where adult RGCs undergo cell death in response to injury (*Bähr, 2000*). Therefore, we compared transcriptomic profiles of *strip1* mutant eyes to those of adult zebrafish RGCs following ONI (*Veldman et al., 2007*) or adult eyes after optic nerve crush (*McCurley and Callard, 2010*). Indeed, there were several genes commonly upregulated in all three models, namely, *jun*, *atf3*, *gap43*, *stmn4*, *sox11b*, and *adcyap1b*

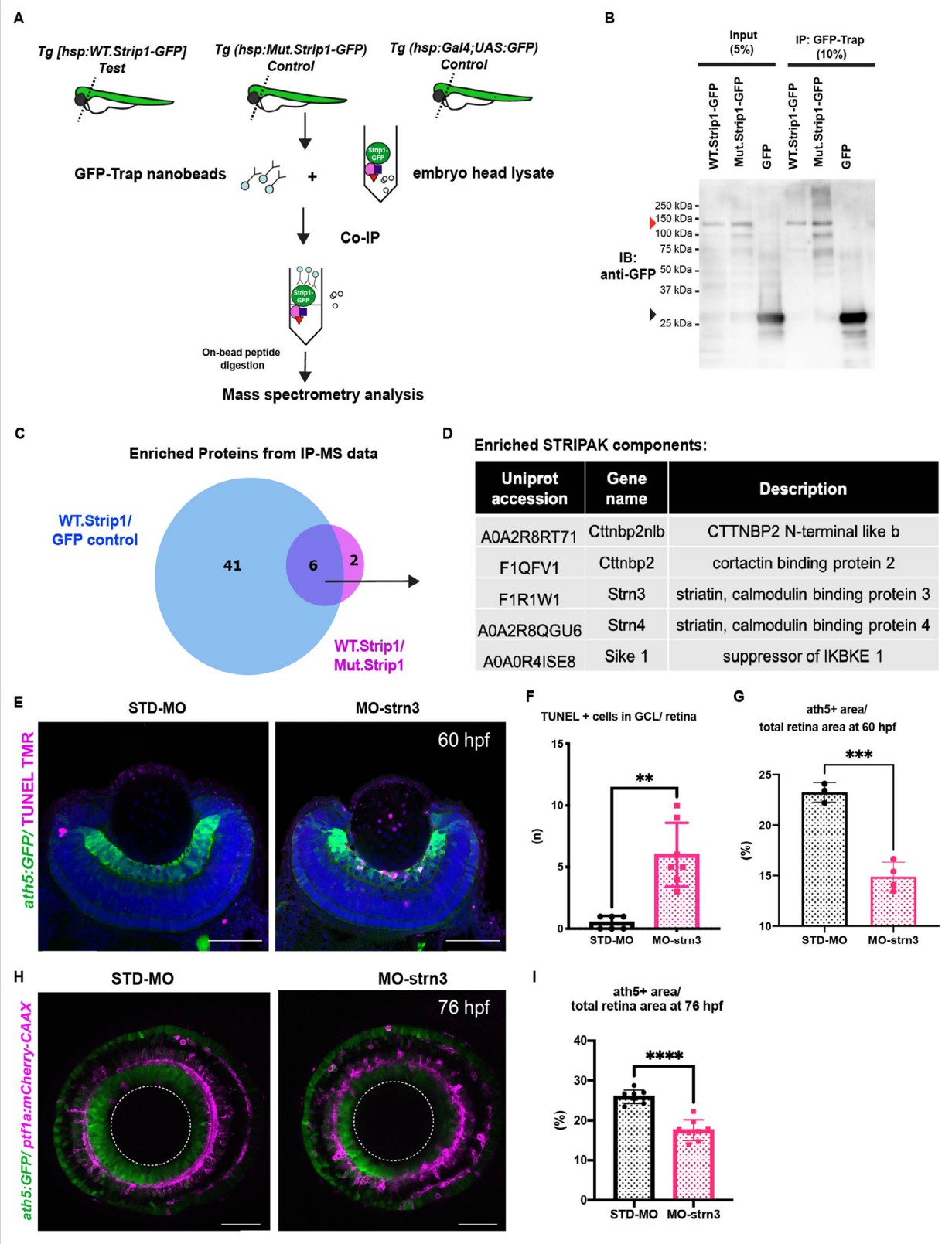

**Figure 5.** Strn3 is a Strip1-interacting partner that promotes retinal ganglion cell (RGC) survival. (**A**) Design of co-immunoprecipitation coupled with mass spectrometry (Co-IP/MS) to identify zebrafish Strip1-interacting partners. Embryos carrying the transgenes *Tg[hsp:WT.Strip1-GFP]*, *Tg[hsp:Mut. Strip1-GFP]*, or *Tg[hsp:Gal4;UAS:GFP]* were used to pull-down wild-type GFP-tagged Strip1, mutant GFP-tagged Strip1 or only GFP, respectively. Head lysates from 2-dpf zebrafish embryos were subjected to immunoprecipitation using GFP-Trap beads. Immunoprecipitates were digested and analyzed

*Figure 5 continued on next page*

*Figure 5 continued*

by mass spectrometry (MS). (**B**) Western blotting of whole head lysates (input) and immunoprecipitates (IP) using anti-GFP antibody. Red and black arrowheads indicate the expected band sizes for Strip1-GFP (120 kDa) and GFP (26 kDa), respectively. (**C**) Venn diagram comparing proteins significantly enriched in WT.Strip1-GFP relative to Control GFP (blue) and WT.Strip1-GFP relative to Mut.Strip1-GFP (magenta). Six proteins are commonly enriched in both groups, FC >2, p < 0.05. n = 3 for WT.Strip1-GFP and Mut. Strip1-GFP and n = 2 for GFP-control. (**D**) Five components of the STRIPAK complex found from six proteins commonly enriched in (**C**). (**E**) Transferase dUTP nick end labeling (TUNEL) of 60-hpf retinas of *Tg[ath5:GFP]* transgenic embryos injected with standard MO and MO-strn3. RGCs and apoptotic cells are labeled with *ath5:GFP* and TUNEL, respectively. Nuclei are stained with Hoechst (blue). (**F**) The number of TUNEL+ cells in ganglion cell layer (GCL). Mann–Whitney *U*-test, n ≥ 6. (**G**) Percentage of ath5+ area relative to total retinal area. Student's *t*-test with Welch's correction, n ≥ 3.(**H**) Confocal images of retinas of 76-hpf *Tg[ath5:GFP; ptf1a:mCherry-CAAX]* transgenic embryos injected with standard MO and MO-strn3. *ath5:GFP* and *ptf1a:mCherry-CAAX* label RGCs and amacrine cells (ACs), respectively. (**I**) Percentage of ath5+ area relative to total retinal area. Student's *t*-test with Welch's correction, n = 8. Scale bar, 50 μm (**E, H**). For all graphs, data are represented as means ± standard deviation (SD). **p < 0.01, ***p < 0.001, and ****p < 0.0001.

The online version of this article includes the following source data and figure supplement(s) for figure 5:

**Source data 1.** Data for *Figure 5B*.

**Source data 2.** Data for *Figure 5C*.

**Source data 3.** Data for *Figure 5F,G,I*.

**Figure supplement 1.** Components of the STRIPAK complex are highly enriched in the interactome of zebrafish Strip1 and their retinal expression, according to published scRNA-seq data.

**Figure supplement 2.** Efficient and specific knockdown of zebrafish Strn3 using morpholinos.

**Figure supplement 2—source data 1.** Data for *Figure 5—figure supplement 2A*.

**Figure supplement 2—source data 2.** Data for *Figure 5—figure supplement 2B*.

---

(*Figure 6—figure supplement 1C*). These findings suggest that Strip1-deficient zebrafish RGCs share a similar stress response with adult RGCs following ONI.

In *strip1^rw147* mutants, *jun* was among the top upregulated stress response markers. Jun (the zebrafish homolog of mammalian c-Jun) is the canonical target of the Jun N-terminal kinase (JNK) pathway, which belongs to the MAPK super family. JNK/c-Jun signaling is a key regulator of stress-induced apoptosis (*Dhanasekaran and Reddy, 2008*; *Ham et al., 2000*). Activation of the JNK pathway involves phosphorylation events that end with c-Jun phosphorylation and transactivation, which in turn activates *c-jun* gene expression (*Eilers et al., 1998*). We stained *strip1^rw147* mutant retinas with anti-phosphorylated c-Jun (p-Jun) antibody. At 54 hpf, *strip1^rw147* mutants showed significantly elevated levels of p-Jun compared to wild-type siblings. This elevation is specifically localized in RGCs visualized with zn5 antibody (*Figure 6C, D*). We confirmed that Jun phosphorylation occurs as early as 48 hpf, when we first observe RGC death (*Figure 6—figure supplement 2A, B*). Interestingly, at 48 hpf, p-Jun localizes in RGCs at the ventronasal patch, which correspond to the earliest-born RGCs. Likewise, p-Jun was significantly elevated in RGCs of *strn3* morphants at 49 hpf compared to control-injected embryos (*Figure 6E, F*). Next, we knocked down Jun using a previously described morpholino (MO-jun) (*Gan et al., 2008*; *Han et al., 2016*). At 60 hpf, apoptosis was significantly inhibited in the GCL of *strip1^rw147* mutants injected with MO-jun, compared to *strip1^rw147* mutants injected with a standard control morpholino (*Figure 6G, H*). Accordingly, at 76 hpf, RGCs were partially but significantly recovered in *strip1^rw147* mutants injected with MO-jun, compared to *strip1^rw147* mutants injected with a standard control morpholino (*Figure 6I and J*). Taken together, Strip1 and Strn3 suppress Jun-mediated apoptotic signaling in RGCs.

## Bcl2 rescues RGC survival in *strip1* mutants, but surviving RGCs do not project their dendrites to the IPL

The anti-apoptotic B-cell lymphoma 2 (Bcl2) is a key regulator of mitochondria-dependent apoptosis in neurons, including RGCs, both during survival and in response to injury (*Anilkumar and Prehn, 2014*; *Bähr, 2000*; *Bonfanti et al., 1996*; *Maes, 2017*). In addition, JNK/c-Jun activation induces neuronal apoptosis by modulating BCL2 family proteins (*Guan et al., 2006*; *Hollville et al., 2019*; *Whitfield et al., 2001*). Thus, we combined *strip1^rw147* mutants with the transgenic line *Tg[hsp:mCherry-Bcl2]*, which overexpresses mCherry-tagged Bcl2 protein under control of a heat shock promoter (*Nishiwaki and Masai, 2020*). Bcl2 overexpression significantly inhibited RGC apoptosis in *strip1^rw147* mutants (*Figure 7A, B*). Accordingly, at 78 hpf, RGCs were partially but significantly recovered in

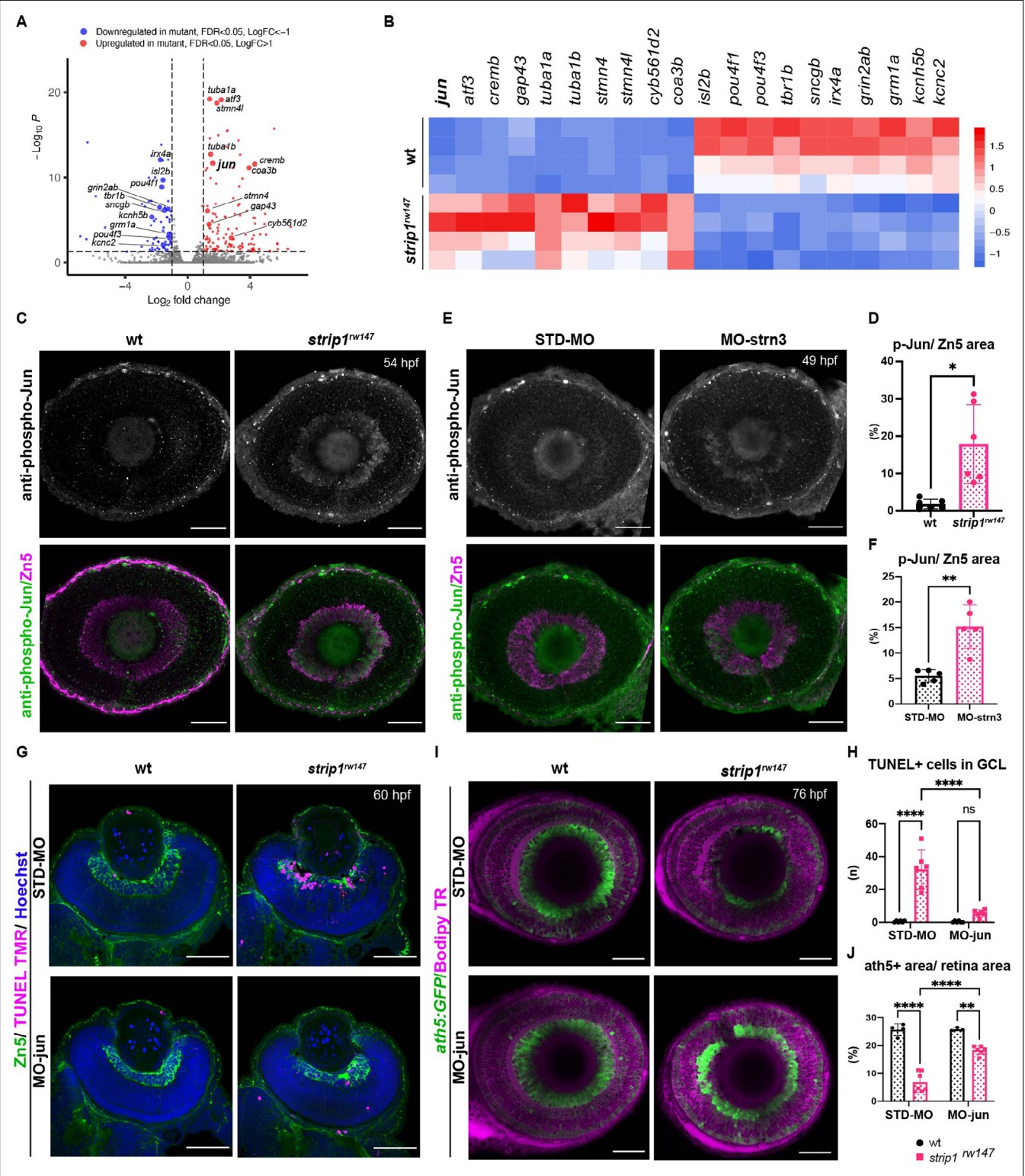

**Figure 6.** Jun is a key mediator of retinal ganglion cell (RGC) death in the absence of Strip1. (**A**) Volcano plot showing differentially expressed genes (DEGs) in *strip1^rw147* mutants compared to wild-type siblings. Colored points represent genes that are significantly upregulated (131 genes, red) or downregulated (75 genes, blue). Data are obtained from four independent collections of 62-hpf embryo eye cups. FDR < 0.05, log_2 FC > |1|. (**B**) Heatmap of expression values (z-score) representing selected DEGs in *strip1^rw147* mutants compared to wild-type siblings. (**C**) Whole-mount labeling of 54-hpf wild-type and *strip1^rw147* mutant retinas with anti-phospho-Jun antibody and zn5 antibody, which label active Jun and RGCs, respectively. (**D**) Percentage of phospho-Jun area relative to zn5 area at 54–58 hpf. Student's *t*-test with Welch's correction, *n* = 6. (**E**) Whole-mount labeling of 49-hpf

*Figure 6 continued on next page*

*Figure 6 continued*

wild-type embryos injected with standard MO or MO-strn3 with anti-phospho-Jun antibody and zn5 antibody, respectively. (**F**) Percentage of phospho-Jun area relative to zn5 area at 49 hpf. Student's *t*-test with Welch's correction, *n* = 5. (**G**) Transferase dUTP nick end labeling (TUNEL) and zn5 antibody labeling of 60-hpf wild-type and *strip1^rw147^* mutant retinas injected with standard MO and MO-Jun. Nuclei are stained with Hoechst. (**H**) The number of TUNEL+ cells in GCL per retina. Two-way analysis of variance (ANOVA) with the Tukey multiple comparison test, *n* = 6. (**I**) Confocal images of 76-hpf wild-type and *strip1^rw147^* mutant retinas injected with standard-MO and MO-Jun. Embryos carry the transgene *Tg[ath5:GFP]* to label RGCs and are stained with bodipy TR methyl ester to visualize retinal layers. (**J**) Percentage of ath5+ area relative to total retinal area. Two-way ANOVA with the Tukey multiple comparison test, *n* ≥ 3. Scale bar, 50 µm (**C, E, G, I**). For all graphs, data are represented as means ± SD. ns, not significant, *p < 0.05, **p < 0.01, and ****p < 0.0001.

The online version of this article includes the following source data and figure supplement(s) for figure 6:

**Source data 1.** Data for *Figure 6A*.

**Source data 2.** Data for *Figure 6D,F,J,H*.

**Figure supplement 1.** Gene ontology (GO) enrichment analysis of differentially expressed genes (DEGs) from RNA-seq of *strip1* mutants and comparison between upregulated DEGs in *strip1* mutant transcriptomes vs. transcriptomes of zebrafish retinal ganglion cells (RGCs) under stress.

**Figure supplement 2.** Jun is activated in retinal ganglion cells (RGCs) of *strip1* mutants at 48 hpf.

**Figure supplement 2—source data 1.** Data for *Figure 6—figure supplement 2B*.

*strip1^rw147^* mutants overexpressing Bcl2, compared to non-transgenic mutants (*Figure 7C, D*). Thus, loss of RGCs in *strip1^rw147^* mutants depends on the mitochondria-mediated apoptotic pathway.

Surprisingly, *strip1^rw147^* mutants overexpressing Bcl2 still displayed IPL defects. In this case, the IPL was not formed at the interface between surviving RGCs and ACs, but instead, a thin IPL-like neuropil was ectopically formed in the middle of presumptive AC layer (*Figure 7C*). Thus, a fraction of presumptive ACs were abnormally located between surviving RGCs and the IPL-like neuropil, although this AC fraction did not intermingle with surviving RGCs (*Figure 7C*, bottom panels, asterisks). In addition, surviving RGCs in *strip1^rw147^* mutants apparently fail to project their dendrites to the IPL-like thin neuropil, consistent with our previous findings on the cell autonomous role of Strip1 in RGC neurite patterning. Upon closer examination, few surviving RGCs successfully innervate the IPL, and such areas show less infiltration of ACs (*Figure 7C*, bottom panels, arrowheads). These data confirm an additional role of Strip1 in dendritic patterning of RGCs, which is likely to prevent ectopic IPL-like neuropil formation in the AC layer.

## Discussion

Over the past decade, Strip1/Strip has emerged as an essential protein in embryonic development (*Bazzi et al., 2017*; *La marca, 2019*; *Neal et al., 2020*; *Sakuma et al., 2014*; *Sakuma et al., 2015*). Using zebrafish, we demonstrate that Strip1 performs multiple functions in development of inner retinal neural circuit. First, we discovered a novel neuroprotective mechanism governed by Strip1, probably through the STRIPAK complex, to suppress Jun-mediated proapoptotic signaling in RGCs during development (*Figure 8A*). In addition, we demonstrate that Strip1-mediated RGC maintenance is essential for laminar positioning of other retinal neurons and structural integrity of the IPL.

RGCs are the most susceptible retinal neurons to cell death, both during development and in response to injury. Unlike zebrafish retina, in which only 1.06% of RGCs die during development (*Biehlmaier et al., 2001*), around 50% of mammalian RGCs undergo apoptotic cell death (*Bähr, 2000*; *Fawcett et al., 1984*). Similarly, the survival rate of mouse RGCs following ONI is only ~8% compared to a survival rate of ~75% of zebrafish RGCs (*Li et al., 2020*; *Zou et al., 2013*). This suggests that RGC survival signals are more active in zebrafish retina. Recently, many studies have been seeking to identify such zebrafish-specific survival mechanisms, with the aim to develop therapy that can prevent death of mammalian RGCs (*Chen et al., 2021*). Interestingly, loss of zebrafish Strip1 causes an elevated apoptotic stress response profile in embryonic retinas having a degree of overlap with adult RGCs post-ONI. Indeed, five out of the six overlapping upregulated markers (*jun*, *atf3*, *stmn4*, *sox11b*, and *adcyap1b*) are commonly upregulated in retinal transcriptomic studies of mammalian ONI models (*Wang et al., 2021*). Jun is the canonical target of JNK signaling and JNK/Jun activation is a major cause of axonal injury- or glaucoma-induced RGC death (*Fernandes et al., 2012*; *Fernandes et al., 2013*; *Syc-Mazurek et al., 2017*). Thus, Jun signaling appears to be a common mediator of RGC death among vertebrates. Our findings will open promising new research avenues to determine

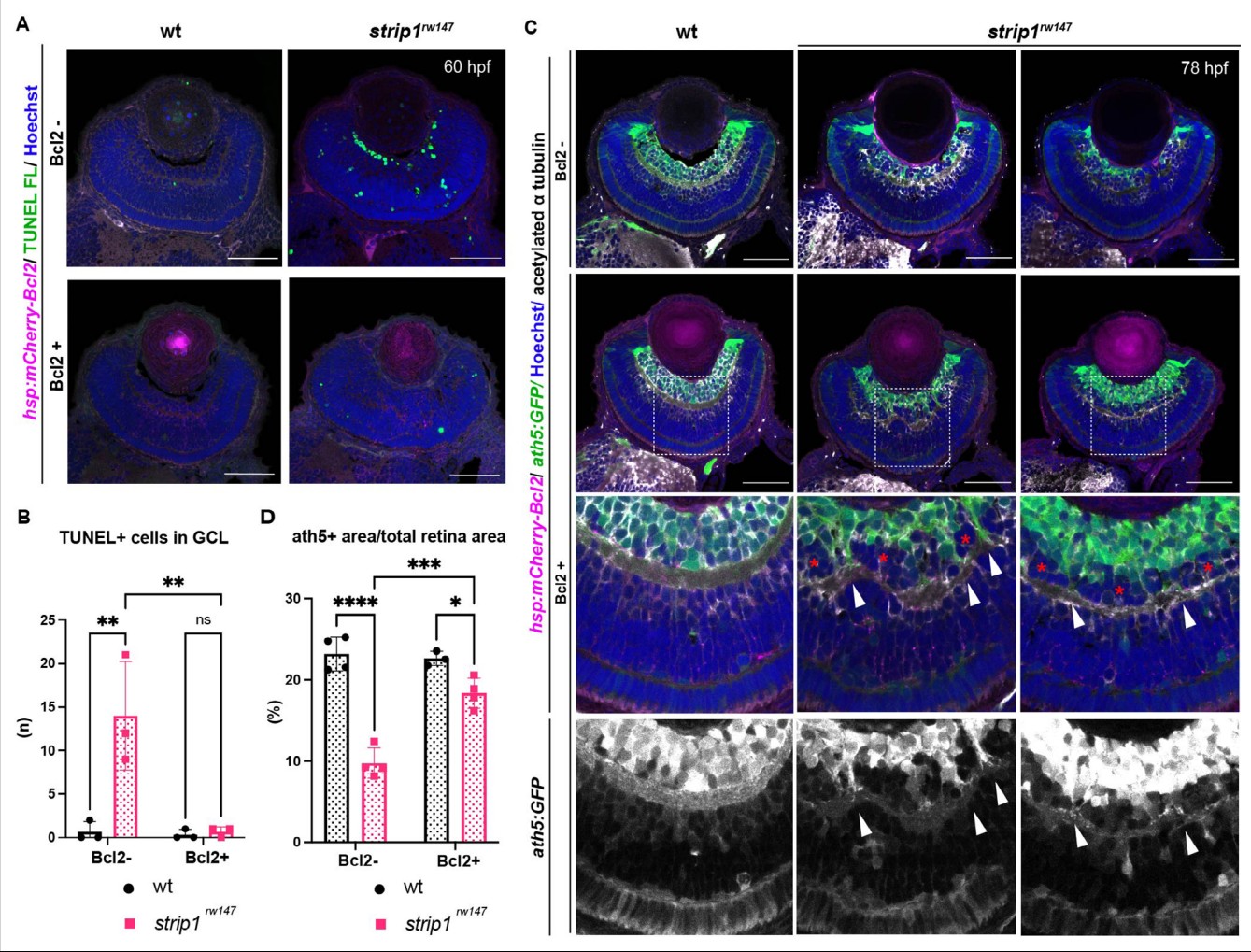

**Figure 7.** Bcl2 rescues retinal ganglion cell (RGC) death in *strip1* mutants, but surviving RGCs do not project their dendrites to the inner plexiform layer (IPL). (**A**) 60-hpf wild-type and *strip1^rw147* mutants combined with the transgenic line *Tg[hsp:mCherry-Bcl2]*. Nontransgenic embryos (Bcl2−, top panels) are compared to transgenic embryos (Bcl2+, bottom panels) after heat shock treatment. Apoptotic cells are visualized by transferase dUTP nick end labeling (TUNEL) FL and fluorescent signals from mCherry-Bcl2 are shown. Nuclei are stained with Hoechst. (**B**) The number of TUNEL+ cells in ganglion cell layer (GCL). Two-way analysis of variance (ANOVA) with the Tukey multiple comparison test, *n* = 3. (**C**) 78-hpf wild-type and *strip1^rw147* mutant retinas combined with the transgenic lines, *Tg[ath5:GFP]* and *Tg[hsp:mCherry-Bcl2]*. Nontransgenic embryos (Bcl2−, top panels) are compared to transgenic embryos (Bcl2+, bottom panels) after heat shock treatment. RGCs are labeled with *ath5:GFP* and fluorescent signals from mCherry-Bcl2 are shown. Anti-acetylated α-tubulin labels the IPL. Nuclei are stained with Hoechst. Arrowheads represent areas where RGC dendrites contribute to the IPL. Asterisks denote areas where RGC dendrites fail to project to the forming IPL and a fraction of presumptive amacrine cells is located between them. (**D**) Percentage of ath5+ area relative to retinal area. Two-way ANOVA with the Tukey multiple comparison test, *n* ≥ 3. Scale bar, 50 μm (**A, C**). For all graphs, data are represented as means ± standard deviation (SD). ns, not significant, *p < 0.05, **p < 0.01, ***p < 0.001, and ****p < 0.0001.

The online version of this article includes the following source data for figure 7:

**Source data 1.** Data for *Figure 7B,D*.

whether Strip1-mediated Jun suppression can modulate proapoptotic signaling in adult RGCs of both zebrafish and higher vertebrates.

Why is Jun activated within RGCs in absence of Strip1? We reported that Strip1 is cell autonomously required for both RGC survival and RGC neurite patterning. This raises questions about whether Jun activation occurs due to failure of RGCs to connect with their pre/postsynaptic partners or whether this activation is connectivity independent. Our data at cellular and molecular levels show that the Jun-mediated apoptotic program starts as early as 48 hpf. On the other hand, previous reports suggest that RGCs start to project apical dendrites and innervate the IPL at around 55–60 hpf, following lamination cues from ACs (*Choi et al., 2010*; *Mumm et al., 2006*). In addition, synaptogenesis in the IPL

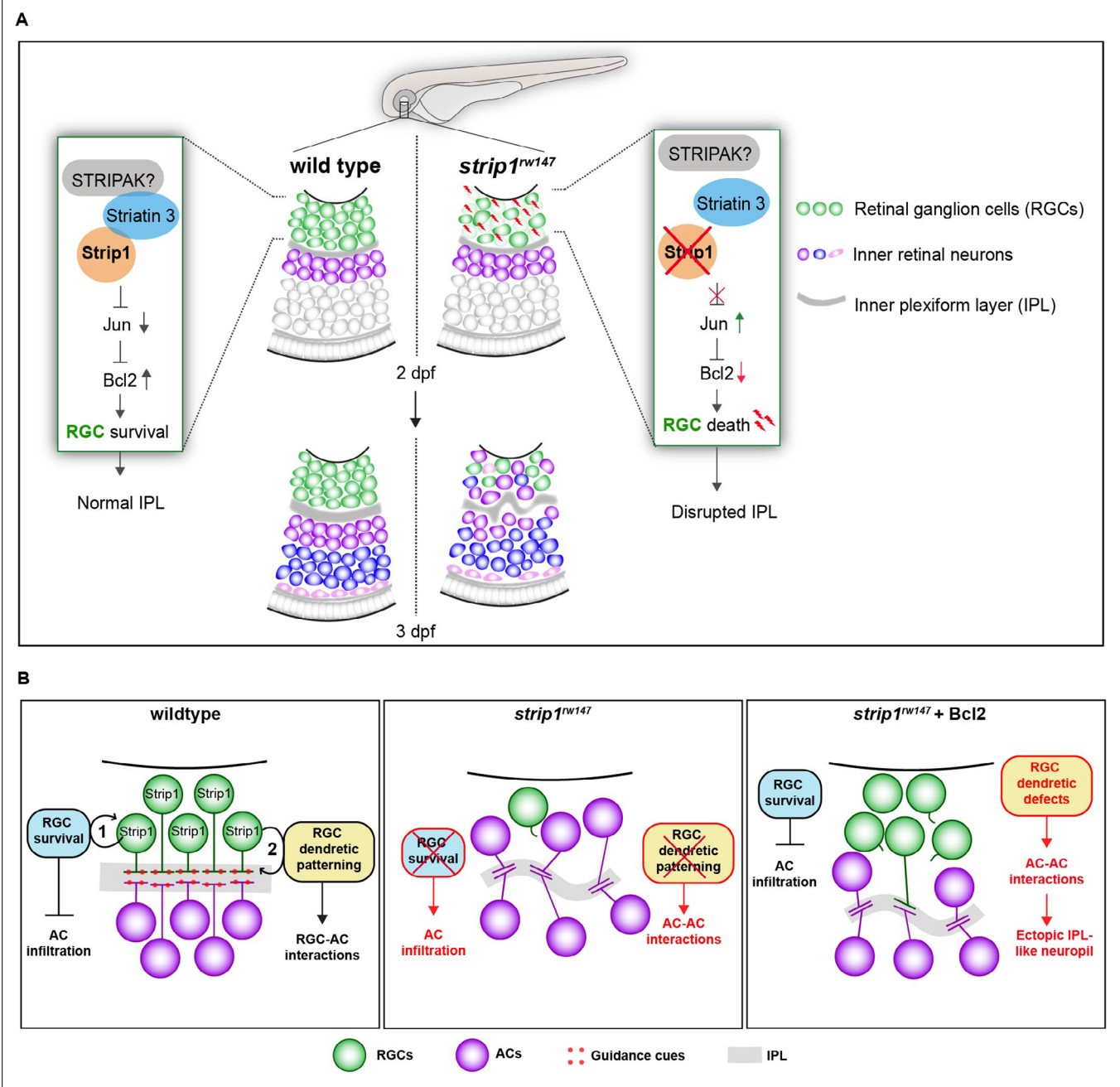

**Figure 8.** Summary of developmental and molecular events that underlie Strip1 function in inner retinal circuit formation. (**A**) In wild-type retina, Strip1 suppresses Jun-mediated proapoptotic signals, probably through the STRIPAK complex, to maintain retinal ganglion cells (RGCs) during development. In the absence of Strip1, Jun is activated in RGCs leading to severe degeneration of RGCs as early as 2 dpf. Subsequently, cells in the inner nuclear layer (INL) abnormally infiltrate the ganglion cell layer (GCL) leading to a disrupted inner plexiform layer (IPL). (**B**) Proposed model for Strip1's role within RGCs to regulate amacrine cell (AC) positioning and IPL formation. In wild type, Strip1 regulates (1) RGC survival to prevent AC infiltration, and (2) RGC dendritic patterning to promote RGC–AC interactions. In *strip1rw147* mutants, both mechanisms are perturbed, leading to AC infiltration, increased AC–AC interactions, and IPL defects. In Bcl2-rescued *strip1rw147* mutants, survived RGCs prevent AC infiltration. However, RGC dendritic defects lead to increased AC–AC interactions and ectopic IPL formation.

starts at around 60 hpf (*Schmitt and Dowling, 1999*). Furthermore, our time-lapse imaging shows that RGC death starts prior to IPL malformation. Therefore, it is unlikely that failure of connectivity in the IPL is the primary cause of RGC death. On the other hand, understanding the contribution of possible connectivity defects in the optic tectum to RGC death is more challenging. In wild-type zebrafish embryos, complete optic nerve transection in 5-dpf larvae does not induce prominent RGC

death (*Harvey et al., 2019*). However, at 48 hpf, we observe that Jun activation starts in the earliest-born retinal neurons, which coincides with the timing when wild-type, early-born RGCs start to innervate the optic tectum (*Burrill and Easter, 1994*; *Stuermer, 1988*). It is possible that in wild-type zebrafish embryos, when connectivity to the optic tectum is compromised, functional Strip1-mediated survival machinery prevents stress-induced RGC apoptosis. However, in *strip1* mutants, this survival machinery is disrupted, leading to RGC death. Future studies can help clarify whether Jun-mediated apoptosis is caused by elongation defects of RGC axons, or by connectivity-independent intrinsic cell death mechanisms.

Although we were unable to determine the direct molecular link that underlies Strip1-mediated Jun suppression, our findings strongly suggest the involvement of the STRIPAK complex in this process. Our proteomic assays revealed that recruitment of many STRIPAK components is compromised in *strip1* mutants. Also, we demonstrate that Strip1 interacts with Strn3, and both Strip1 and Strn3 show overlapping roles in RGC survival. Recently, several studies on the human STRIPAK complex found that STRIP1 and STRN3 are organizing centers for the STRIPAK complex and that their mutant forms compromise complex assembly and function (*Jeong et al., 2021*; *Tang et al., 2019*). Modulation of JNK/Jun signaling by the STRIPAK complex is supported by several studies. MAP4 kinases activate the JNK signaling pathway and they are among kinase family members that are recruited and dephosphorylated by the STRIPAK complex (*Fuller et al., 2021*; *Hwang and Pallas, 2014*; *Kim et al., 2020*; *Seo et al., 2020*). Moreover, JNK signaling is activated in STRIP1/2-knockout human cell lines (*Chen et al., 2019*). Similarly, the interaction between Strip and CKa (*Drosophila* homolog of Striatins) suppresses JNK signaling in *Drosophila* testis (*La marca, 2019*). Thus, it is likely that Strip1 and Strn3 function in the context of the STRIPAK complex to modulate JNK/Jun activity, thereby promoting RGC survival. To our knowledge, this study is the first in vivo evidence for a functional interaction between STRIPAK components and Jun signaling in vertebrates.

There are still many gaps in our knowledge of mechanisms underlying IPL development. Which molecular cues dictate the laminar positioning of inner retinal neurons? Which cell types are essential for IPL formation? It has been proposed that IPL development is a robust process. Upon genetic elimination of different inner retinal cells, the remaining cells manage to form an IPL-like neuropil (*Randlett et al., 2013*). However, it is widely agreed that ACs play the dominant role in IPL initiation. Elegant time-lapse experiments show that ACs project their neurites to form a proto-IPL (*Chow et al., 2015*; *Godinho et al., 2005*). This presumed IPL guides RGCs to extend dendritic arbors and stratify, although they are born earlier than ACs (*Mumm et al., 2006*). Other studies propose an active role for RGCs in shaping the developing IPL (*Kay et al., 2004*). We found that RGC death in *strip1* mutants is strongly linked to abnormal infiltration of ACs and BPs in the GCL and the perturbed IPL. This coincides with the phenotype of the *lakritz* mutant, in which similar defects occur when RGC genesis is inhibited (*Kay et al., 2001*; *Kay et al., 2004*). In knockout mice in which atypical Cadherin Fat3 is absent in both RGCs and ACs, ACs invade the GCL abnormally. However, AC-specific knockout mice do not exhibit such positioning defects (*Deans et al., 2011*). Therefore, we reintroduce a model proposed by *Kay et al., 2004*, in which both RGCs and ACs play distinct roles in shaping the developing IPL. In this model, RGCs provide positional cues for migrating ACs to initiate a proper IPL program, whereas ACs subsequently project their dendritic plexuses to establish the foundation for a proto-IPL.

So far, the RGC-dependent mechanisms that instruct the laminar positioning of ACs and IPL development remain unknown. However, Bcl2-rescued *strip1* mutants provide valuable new insights into such mechanisms. Bcl2-rescued mutants show defects in dendritic patterns of surviving RGCs, which are associated with an ectopic IPL formed amidst presumptive ACs. Thus, we propose that RGCs serve dual functions in IPL development (summarized in *Figure 8B*): (1) RGCs act as a physical barrier that prevents abnormal infiltration of ACs into the GCL and (2) RGCs show dendritic guidance cues that establish interactions between RGCs and ACs for a proper IPL program. It is unclear what guidance cues participate in this process. Possible candidates are N-cadherin and Semaphorin-3 receptors, Neuropillin-1 (Nrp1) and PlexinA1. A hypomorphic allele of zebrafish *n-cadherin* mutants compromises IPL formation with abnormal neurite patterning of INL cells (*Masai et al., 2003*). In *Xenopus*, Nrp1 and PlexinA1 inhibition induced randomly oriented dendritic patterning of RGCs, similar to zebrafish *strip1* mutants (*Kita et al., 2013*). Future experiments will clarify whether these molecules participate in communication among RGCs and ACs to establish a proper IPL. Lastly, we show that Strip1 is required for proper neurite patterning of RGCs, and probably small subsets of ACs and BPs.

This is supported by established roles of *Drosophila* Strip in dendritic branching and axon elongation (*Sakuma et al., 2014*). Future studies on cell-specific Strip1 knockout models could clarify Strip1 function in retinal neurite morphogenesis.

In summary, we demonstrate that a series of Strip1-mediated regulatory mechanisms constructs retinal neural circuit through RGC survival and neurite patterning of retinal neurons. Our findings provide valuable insights to mechanistic understanding of JNK/Jun-mediated apoptotic pathway and correct assembly of synaptic neural circuits in the brain. For medical perspective, our findings on a similar stress response between zebrafish *strip1* mutants and mammalian ONI models pave the way for future research on potential Strip1-mediated therapeutic targets that could help mitigate RGC degeneration in glaucoma and optic neuropathies.

## Materials and methods

### Transgenic fish lines

To visualize RGCs, the transgenic line *Tg[ath5:GFP]$^{rw021}$* (*Masai et al., 2003*) was used. In this line, GFP is expressed under control of the *ath5* (also referred to as *atoh7*) promoter. The transgenic lines *Tg(Gal4-VP16,UAS:EGFP)xfz43* or *xfz43* and *Tg(Gal4-VP16,UAS:EGFP)xfz3* or *xfz3* (*Zhao et al., 2009*) are enhancer trap lines and were both used to label distinct populations of BPs. To visualize ACs, the transgenic line *Tg[Ptf1a:mCherry-CAAX]$^{oki067}$* was generated by injecting the plasmid *pTol2[ptf1a:mCherry-CAAX]* into one-cell-stage fertilized eggs, together with Tol2 transposase mRNA.

For Bcl2 overexpression experiments, the line *Tg[hs:mCherry-tagged Bcl2]$^{oki029}$* (*Nishiwaki and Masai, 2020*) was employed, in which mCherry-tagged Bcl2 at the N-terminus is overexpressed under control of the heat shock promoter. For overexpression of wild-type and *rw147* mutant forms of zebrafish Strip1, the lines *Tg[hsp:WT.Strip1-GFP]$^{oki068}$* and *Tg[hsp:Mut.Strip1:GFP]$^{oki069}$* were generated, respectively, to express GFP-tagged Strip1 at the C-terminus under control of the heat shock promoter. For line generation, the DNA constructs *pTol2[hsp:WT.Strip1-GFP]* and *pTol2[hsp:Mut.Strip1-GFP]* were injected into one-cell-stage fertilized eggs together with Tol2 transposase mRNA. These injected F0 embryos were bred up to the adult stage and used to identify founder fish that produce F1 generation embryos showing stable GFP expression. Transgenic lines were established in the F2 generation. The transgenic line *Tg[hsp:Gal4;UAS:EGFP]* was generated by combining *Tg[hsp:gal4]$^{kca4}$* (*Scheer et al., 2002*) with *Tg[UAS:EGFP]* (*Köster and Fraser, 2001*) to express EGFP under control of the heat shock promoter, while *Tg[UAS:MYFP]* expresses EYFP fused to the membrane targeting palmitoylation signal of gap43 under the control of the 14XUAS E1b promoter (*Schroeter et al., 2006*). Some of the mentioned transgenic lines were combined with the mutant line *strip1$^{rw147}$*. The steps of plasmid construction are described below in detail.

### Mutant line generation, mutant identification, and genotyping

The *strip1$^{rw147}$* mutant line was generated from a mutagenesis screen (*Masai et al., 2003*) that used RIKEN Wako (RW) as a wild-type strain. Mutation mapping and subsequent experiments were carried out in the genetic background of WIK and Okinawa wild type (oki), respectively. The *rw147* mutation was mapped on a genomic region in chromosome 22 flanked by two self-designed polymorphic markers; AL928817-12: (5′-TTCAACATCTGCTTTTCCTCCT-3′ and 5′-TCATGTCCCAGAAATCACAC AT-3′) and zk253D23-4 (5′-CATTCTTCATTAAAGAGATCAGTGTGA-3′ and 5′-AGTGATCACACACCCC CACT-3′). In addition, the location of the *rw147* mutation was further restricted using another self-designed polymorphic marker Zk286J17-3 (5′- TTCACATTTACATTTTTCTGAACATTT-3′ and 5′-CACA CAGCCTTCTCTTGCAC-3′) as no recombination was detected.

From 3 dpf, *strip1$^{rw147}$* homozygous mutants (*strip1$^{-/-}$*) are distinguished from wild-type siblings (*strip1$^{+/+}$* or *strip1$^{+/-}$*) by external morphology since they exhibit cardiac edema, an abnormal lower jaw, and smaller eyes. From 54 to 72 hpf, *strip1$^{rw147}$* homozygous mutants are screened with Acridine Orange (AO) live staining to detect apoptotic cells (*Casano et al., 2016*) or Bodipy TR live staining to visualize lamination defects (*Choi et al., 2010*). Prior to 54 hpf, genotyping of *strip1$^{rw147}$* mutant embryos was performed by sequencing. The primer set, 5′-CGTGTGTTTTCAGGGTGTT-3′ and 5′-TCACCATCCCAAACAGCATA-3′, was used. The 257 bp PCR amplicons were amplified using Phusion Hot Start II (Thermo Fisher Scientific) and sequenced for genotyping.

The *strip1*^crisprΔ10 (officially referred to as *strip1*^oki8) mutant line was generated using CRISPR-Cas9 gene editing technology. The gRNA sequence, 5′-CCCGCGTCCGCCTCTGACCTCAT-3′, was designed using chopchop (https://chopchop.cbu.uib.no) and it targets exon 9 of *strip1* gene. One-cell-stage embryos were injected with 200 ng/μl gRNA and 500 ng/μl Cas9 protein (FASMAC). F1 mutant founders were identified by sequencing. A 10-bp deletion was introduced at nucleotide 932 of the Strip1 coding sequence, resulting in a frameshift at amino acid 313 and a premature stop at amino acid 330. The primer set 5′-CGTTCCAAATCATTGAAACAGA-3′ and 5′-TGTTTGTGATGTGTTGACCT TG-3′ was used for genotyping. PCR amplicons were run on 15% polyacrylamide gels for identification of wild-type siblings and mutants.

All generated transgenic and mutant lines were combined with the zebrafish pigmentation mutant, *roy orbison (roy)* (**D'Agati et al., 2017**) to remove iridophores and enhance live imaging.

## Molecular cloning

To generate pTol2[hsp:WT.Strip1-GFP] and pTol2[hsp:Mut.Strip1-GFP], a PCR strategy was used to amplify ~2.5 kb Strip1 cDNA from total cDNA of 4-dpf wild-type and *strip1*^rw147 zebrafish embryos using the primers 5′-AGACTTGTGTCAGCGTGACGCGAG-3′ and 5′- ACTCTAGCAAGTGTAGTGTT GTTGATG-3′. Then, using a Gibson Assembly Cloning Kit, a *strip1* cDNA fragment was cloned into a Tol2 transposon vector pT2AL200R150G (**Urasaki et al., 2006**) at the *XhoI* and *ClaI* sites with a heat-shock inducible promoter (hsp) at the N-terminus (**Halloran et al., 2000**) and a GFP tag at the C-terminus (separated by a linker sequence, CTCGAGGGAGGTGGAGGT). For pTol2[ptf1a:mCherry-CAAX] construction, pG1[ptf1a:GFP] was used as donor plasmid, which was kindly provided by the Francesco Argenton lab. A 5.5-kb fragment of the *ptf1a* promoter sequence was retrieved at *HindIII* and *SmaI* sites and inserted into a pBluescript SK (+) (Stratagene) shuttle vector upstream of the membrane-targeting mCherry-CAAX sequence. Then, the ptf1a:mCherry-CAAX sequence was inserted into the *XhoI and BglII* sites of pT2AL200R150G. The pB[ath5:Gal4-VP16] plasmid was constructed by inserting a 6.6-kb fragment of the *ath5* 5′-enhancer/promoter region (including the 5′ UTR) into the *BamHI* site of the pB[Gal4-VP16] plasmid provided by Dr. R. Köster (**Köster and Fraser, 2001**).

## In vivo cell labeling

Single-cell mosaic labeling to visualize RGC morphology was done by injecting 20 ng/μl of pB[ath5:Gal4] into one-cell-stage embryos from intercrosses of *strip1*^rw147 heterozygous fish combined with *Tg[UAS:MYFP]* (**Schroeter et al., 2006**). Likewise, pZNYX-Gal4VP16, a kind gift from the Rachel Wong Laboratory, was injected to visualize ON-BPs (**Schroeter et al., 2006**). Single AC labeling was performed by injecting the DNA construct pG1[ptf1a:GFP] into one-cell-stage embryos from inter-crosses of *strip1*^rw147 heterozygous fish at a concentration of 20 ng/μl (**Jusuf et al., 2012**).

## Morpholino knockdown assay

Embryos produced by intercrosses of wild-type or *strip1*^rw147 heterozygous fish were injected with anti-sense morpholino oligonucleotides at one-cell stage. MO-strip1, MO-strn3, and MO-ath5 (**Pittman et al., 2008**; **Ranawat and Masai, 2021**) were injected at a concentration of 250 μM, whereas MO-jun (**Gan et al., 2008**; **Han et al., 2016**) was injected at a concentration of 125 μM. For each morpholino experiment, the same concentration of the standard control morpholino (STD-MO) was used as a negative control. Detailed morpholino sequences are listed in the Key resources table.

## DiI/DiO injections

To trace the RGC axon projections into the optic tectum, 3-dpf embryos were fixed in 4% paraformaldehyde (PFA) and after washing with phosphate-buffered saline several times, were injected with 2 mg/ml of the lipophilic dyes, DiI and DiO, in the area between the lens and retina. Large injection volumes were applied to label all RGCs. Embryos were incubated overnight at 4°C, and then mounted in 75% glycerol for confocal imaging.

## Histological methods

Zebrafish embryos were embedded for JB4 plastic sectioning and toluidine blue counterstaining, as previously described (**Sullivan-Brown et al., 2011**). Immunolabeling of cryosections and paraffin sections (for anti-Strip1, anti-Pax6, and anti-Prox1 staining) was carried out according to standard

protocols (*Imai et al., 2010*; *Masai et al., 2003*). An antigen retrieval step was performed on paraffin sections by heating in 10 mM citrate buffer, pH 6.0 for 5 min at 121°C. TUNEL was performed using an In Situ Cell Death Detection Kit (Roche) according to the manufacturer's protocol. Nuclear staining was carried out using 1 nM TOPRO3 or 1 ng/ml Hoechst 33342.

Antibodies used in this study and their dilutions are as follows: anti-acetylated α-tubulin (1:1000), anti-Pax6 and anti-Prox1 (1:500), anti-PCNA (1:200), anti-GS (1:150), antibodies against zpr1 and zpr3 (1:100), zn5 antibody (1:50), anti-parvalbumin (1:500), and anti-p-Jun (1:100). Antibody against the peptide sequence of zebrafish Strip1 (amino acids 344–362: EKDPYKADDSHEDEEENDD) was generated using a synthetic peptide and used for immunostaining at 1:1000. For adsorption control, purified antibody was preincubated with 3.6 μg/ml of corresponding blocking peptide for 1 hr at room temperature. Secondary antibodies employed in this study were: Alexa488, 546, and 647 fluorophore-conjugated secondary antibodies used at a concentration of 1:500.

For whole-mount immunostaining against acetylated α-tubulin, 3-dpf embryos were fixed at room temperature for 3 hr in 2% trichloroacetic acid (TCA). Then, embryos were washed in PBTr (*Westerfield, 1995*) (PO$_4$ buffer [0.1 M, pH 7.3] + 0.1% Triton X-100) followed by permeabilization in 0.2% trypsin for 4 min at 4°C. After washing, a post-fixation step in 4% PFA for 5 min at 4°C was applied. Next, blocking was done in 10% goat serum in PBTr for 1 hr at room temperature followed by incubation in mouse anti-acetylated α-tubulin in 1% goat serum/PBTr overnight at 4°C. After washing, embryos were incubated in secondary antibody diluted in 1% goat serum in PBTr overnight at 4°C. Whole-mount immunostaining against p-Jun was performed following standard protocols (*Ungos et al., 2003*). After staining, embryos were mounted in 75% glycerol for confocal imaging.

Whole-mount, in situ hybridization was performed on wild-type zebrafish embryos at specific developmental stages, as previously described (*Xu et al., 1994*). Hybridization was performed overnight at 65°C using *strip1* RNA probe at the concentration 2.5 ng/μl in hybridization buffer. *strip1* probe synthesis was performed according to standard protocols (*Thisse and Thisse, 2008*). Template regions were amplified from *strip1* cDNA using the primers, 5′-AATGCTGCCGAATAAAATGCGAG-3′ and 5′- CCCAGAGTGAACAGGATGCTCT-3′. Antisense and sense probes were synthesized by in vitro transcription using a DIG RNA Labeling Kit (Roche). Following labeling, whole embryos were mounted in 75% glycerol for imaging. To visualize expression patterns in the retina, cryosections were prepared from whole-mount embryos posthybridization.

## Live staining

To visualize lamination patterns, live staining of retinal landmarks was performed by incubating live zebrafish embryos in 100 nM solution of Bodipy TR methyl ester (Thermo Fisher Scientific) in E3 embryo rearing media for 1 hr at room temperature following the manufacturer's protocol. To examine DNA condensation of apoptotic cells in the GCL, zebrafish embryos were incubated for 30 min in 5 μg/ml of AO stain dissolved in egg water. Following staining, embryos were extensively washed with egg water and observed using epifluorescence or imaged using confocal microscopy.

## Overexpression experiments

For rescue experiments, the wild-type form of Strip1, the *rw147* mutant form of Strip1 or Bcl2 was overexpressed in *strip1$^{rw14}$* mutants by heat shock treatment using the transgenic lines *Tg[hsp:WT. Strip1-GFP]*, *Tg[hsp:Mut.Strip1-GFP]*, and *Tg[hs:mCherry-tagged Bcl2]*, respectively. To perform heat shock, embryos from heterozygous intercrosses were incubated for 1 hr at 39°C starting from 27 to 30 hpf and applied every 12 hr until the designated timepoints. For Co-IP/MS, heat shock was applied to embryos from intercrosses of wild-type zebrafish combined with *Tg[hsp:WT.Strip1-GFP]*, *Tg[hsp:Mut. Strip1-GFP]*, or *Tg[hsp:Gal4;UAS:EGFP]*. After screening for transgenic embryos, embryos were either fixed in 4% PFA for histological assays or processed for protein extraction.

## Cell transplantation assays

Single-cell transplantation was performed at blastula stage, as previously described (*Kemp et al., 2009*). Genotypes of donor and host embryos were determined at 3–4 dpf based on morphological phenotype or they were genotyped at earlier time points by sequencing or AO live staining of apoptotic cells. To trace transplanted donor cells in host retinas, 2–5% lysine-fixable dextran rhodamine, Alexa-488 dextran, Alexa-647 dextran or cascade blue dextran were injected in one- to two-cell-stage

donor embryos, depending on study design. To assess the cell autonomy of Strip1 in RGC death, donor embryos from intercrosses of *strip1rw147* heterozygous fish were transplanted into wild-type host embryos. Host embryos with successful retinal transplants were fixed in 4% PFA at 60 hpf and processed for TUNEL. To assess the cell autonomy of Strip1 in RGC dendritic patterning, donor embryos from intercrosses of *strip1rw147* heterozygous fish combined with *Tg[ath5:GFP]* were transplanted into wild-type host embryos. Live imaging of wild-type host retinas was done at 57–58 hpf. To assess the cell autonomy of Strip1 in AC or BP development, donor embryos from intercrosses of *strip1rw147* heterozygous fish combined with *Tg[ptf1a:mCherry-CAAX]* or *Tg[Gal4-VP16,UAS:EGFP] xfz3/xfz43* were transplanted into embryos from intercrosses of *strip1rw147* heterozygous fish. Live imaging of host retinas with successful transplants was done at 3–4 dpf to assess the morphology of donor ACs labeled with mCherry or donor BPs labeled with EGFP. To visualize the retinal lamination phenotype, some hosts were stained with Bopidy TR live stain prior to imaging.

## Microscopy

Imaging of toluidine blue-stained sections and retinal sections following in situ hybridization was performed using a Zeiss upright Axioplan2 equipped with an AxioCam HRC camera, while imaging of whole-mount in situ hybridization embryos was done using a Keyence BZ-X700. An inverted Zeiss LSM 780 was used to scan immunostained retinal sections with a ×40/1.40 Plan-Apochromat Oil objective and whole-mount immunostained embryos using a ×40/1.0 Plan-Apochromat water objective. Glycerol-mounted embryos were placed on glass-bottom depression slides for scanning. For live imaging, zebrafish embryos were anesthetized using 0.02% tricaine (3-amino benzoic acid ethyl ester) dissolved in E3 embryonic medium and mounted laterally in 1% low-melting agarose. Image acquisition of embryo retinas was carried out using an upright Zeiss LSM 710 with a ×40/1.0 W Plan-Apochromat objective or an upright Fluoview FV3000 (Olympus) confocal microscope with a ×40/0.8 water immersion objective.

To perform time-lapse imaging of retinal development, several embryos from intercrosses of *strip1rw147* heterozygous fish carrying the transgenes *Tg[ath5:GFP;ptf1a:mCherry-CAAX]* were mounted simultaneously in a culture dish covered with E3 embryonic medium containing 0.003% PTU and 0.02% tricaine and overlayed with a thin layer of mineral oil to prevent evaporation of E3 medium to minimize embryotoxicity. Retinal z-stacks were acquired consecutively in 1 μm steps every ~2 hr, starting at 48 hpf with undetermined genotypes. Scanning was done using the Multi Area Time Lapse (MATL) Software module of the FV3000 (Olympus) confocal microscope and a motorized XYZ-rotation stage.

All images were processed using ImageJ (NIH, v2.1.0/1.53 C), Imaris (Bitplane, v9.1.2) and Adobe Illustrator software. 3D rendering and analysis of time-lapse movies were performed on Imaris software. Whenever necessary, brightness and contrast display levels for the whole image were adjusted to aid visualization or decrease background noise.

## Western blotting and Co-IP

Heads of noninjected, MO-strip1, STD-MO, and MO-strn3-injected wild-type embryos were dissected at 2 dpf in Leibovitz's L-15 ice cold medium and homogenized in lysis buffer (125 mM NaCl, 50 mM Tris [pH 7.5], 0.5 mM ethylenediaminetetraacetic acid (EDTA) [pH 8], 1% Triton X-100 and 1× cocktail protease inhibitors). Lysates were clarified by centrifugation at 10,000 × *g* for 10 min at 4°C. Equal amounts of denatured clarified lysates were run on 10% Mini-PROTEAN TGX gels for sodium dodecyl sulfate–polyacrylamide gel electrophoresis and transferred to polyvinylidene difluoride (PVDF) membranes using Trans-Blot Turbo PVDF Transfer system. After blocking with 5% skim milk in 0.1% Tween-20 in TBS, immunoblotting was performed using anti-Strip1 (1:500), anti-Strn3 (1:1000) and anti-β-actin (1:5000). HRP-linked rabbit/mouse IgG was used as a secondary antibody. Chemiluminescence signals were detected using a FUJI Las 4000 luminescence image analyzer.

For Co-IP, wild-type embryos carrying the transgenes *Tg[hsp:WT.Strip1-GFP], Tg[hsp:Mut. Strip1-GFP]* or *Tg[hsp:Gal4;UAS:GFP]* were exposed to heat shock starting at 27 hpf, and applied every 12 hr. At 2 dpf, lysates for each biological replicate were prepared from a pool of around 150 embryo heads in NP-40-based lysis buffer (150 mM NaCl, 10 mM Tris [pH 7.5], 0.5 mM EDTA [pH 8], 0.5% NP-40 and 1× cocktail protease inhibitors), as described above. Immunoprecipitation was performed on clarified lysates using anti-GFP (GFP-Trap agarose beads, Chromotek) according to the

manufacturer's protocol. Briefly, clarified lysates were diluted in wash buffer (150 mM NaCl, 10 mM Tris [pH 7.5], 0.5 mM EDTA [pH 8] and 1× cocktail protease inhibitors) to reach 0.1% NP-40. Then, lysates were incubated with pre-equilibrated GFP-Trap beads for 1 hr at 4°C. Afterward, beads were collected by centrifugation and washed in wash buffer five times. To confirm that GFP-fused proteins were successfully pulled down, proteins were eluted from beads by boiling in 1× sample buffer for 5 min. Then, 5% of pre-pulldown lysate (input) and 10% of the pulled-down proteins were run for western blotting as described above using anti-GFP (1:500).

## MS and data analysis

To prepare protein samples for MS analysis, immunoprecipitated protein complexes were eluted from GFP-Trap beads using an on-bead trypsin-based digestion protocol according to the manufacturer's protocol. Digestion was performed overnight at 32°C and under rotation at 400 rpm. Thereafter, digested peptides were cleaned and desalted using $C_{18}$ stage tips, as previously described (*Rappsilber et al., 2007*). Eluted peptides were vacuum-dried and reconstituted in 1% acetic acid, 0.5% formic acid for MS analysis using an Orbitrap-Fusion Lumos mass spectrometer coupled to a Waters nanoACQUITY Liquid Chromatography System. Samples were trapped on a nanoACQUITY UPLC 2 G-V/M Trap 5 µm Symmetry $C_{18}$, 180 µm × 20 mm column and analytical separation was performed on a nanoACQUITY UPLC HSS T3 1.8 µm, 75 µm × 150 mm column. Peptides were fractionated over a 60 min gradient from 1% to 32% acetonitrile with 0.1% formic acid. Solvent flow rate was 500 nl/min and column temperature was 40°C.

LC–MS raw data files were analyzed using Proteome Discoverer (PD, v.2.2, Thermo Fisher Scientific). The SEQUEST algorithm was used to match MS data to the *Danio rerio* (zebrafish) database downloaded from UniProt (July 2021) and the common Repository of Adventitious Protein (cRAP, https://www.thegpm.org/crap). Database search parameters included carbamidomethylation of cysteine as a fixed modification and oxidation of methionine, deamidation of glutamine and asparagine as dynamic modifications. Trypsin was specified as a cleavage enzyme with up to two missed cleavages. Normalization was performed based on specific protein amount (trypsin) and proteins were filtered based on a false discovery rate of $q < 0.05$. Abundance ratios were generated for wild-type compared to mutant and wild-type compared to GFP control. Enriched proteins with abundance ratios ≥2 and adjusted p values <0.05 were considered statistically significant. The Search Tool for the Retrieval of Interacting Genes/Proteins (STRING, v11.0) was used to visualize the Strip1-interaction network with the enriched proteins and to calculate protein–protein interaction values (*Szklarczyk et al., 2017*).

## RNA sequencing and analysis

Total RNA was isolated from four independent biological replicates of 60–64 hpf wild-type siblings and *strip*[rw147] mutant eye cups using a PicoPure RNA Isolation Kit (Thermo Fisher Scientific) according to the manufacturer's instructions. Each biological replicate represented a pool of eye cups obtained from 20 to 30 embryos. All samples had RNA integrity number (RIN) values greater than 8.5. Purified RNA was used for Poly(A)-selected mRNA library preparation with a NEBNext Ultra II Directional RNA Library Prep Kit for Illumina and sequenced on a NovaSeq6000 SP to generate 150 bp paired-end reads. Sequencing reads were quality checked using FastQC (*Andrews, 2010*) and trimmed with FastP (*Chen et al., 2018*). The resulting reads were mapped using hisat v2.1.0 (*Kim et al., 2015*) to the zebrafish reference genome (GRCz11). Mapped reads were counted using featureCounts, v1.6.2 (*Liao, 2014*) and differential gene expression analysis (Mutant vs. Wildtype) was carried out on the counts files using the EdgeR package, v3.32.1 (*Robinson et al., 2010*) in RStudio, v1.4.1106 (*TEAM R, 2016*). Genes with FDR < 0.05 and $\log_2$ FC > |1| were considered statistically significant. Enhanced-Volcano, v1.8.0 (*Blighe et al., 2019*) and pheatmap, v 1.0.12 (*Kolde, 2012*) were used to generate the volcano plot and heatmap, respectively. Gene ontology analyses were performed using Metascape with *D. rerio* as the input species and *M. musculus* as the analysis species (*Zhou et al., 2019*). To analyze published scRNA sequencing data of zebrafish retina at 48 hpf, raw count matrices were analyzed with the Seurat package, v4.0.1 (*Satija et al., 2015*), as previously described (*Xu et al., 2020*). Clustering results were visualized using Uniform Manifold Approximation and Projection (UMAP). The BioVenn web application was used to generate Venn diagrams to compare upregulated differentially expressed genes (DEGs) in this study with published upregulated DEGs in ONI models (*Hulsen et al., 2008*).

## Quantification and statistical analysis

To quantify RGC area, masks were generated for areas with strong ath5+ signals in the retina and quantified using the Color Threshold tool in ImageJ (*Schneider et al., 2012*). Afterward, retinal outlines were defined using the lasso tool and retinal areas were calculated. Data were represented as the percentage of ath5+ area to total retinal area. To calculate the area of p-Jun signals in RGCs, a region of interest (ROI) containing the GCL was defined based on zn5 antibody signal (antigen: alcama, previously referred to as DM-GRASP) to exclude noise at the retina boundary. Areas of p-Jun and zn5 signals were calculated as described above and data were represented as the percentage of p-Jun+ area to zn5+ area.

To quantify apoptotic cells, the number of TUNEL+ cells in GCL or retina was calculated manually within a single retinal section. To quantify the number of ptf1a+ cells that contribute to IPL formation (ACs), cells were manually counted in a unified area (8500 $\mu m^2$) across all samples. Ptf1a+ cells that contributed to the OPL (presumably HCs) were excluded from quantification. To quantify the migration patterns of ptf1a+ cells that contribute to the IPL, cells located at the apical side relative to the IPL were assigned INL+, while cells located at the basal side of the IPL or near the lens were assigned GCL+. To determine the distribution of ptf1a+ cells within transplanted columns, ptf1a+ cells that contributed to the IPL were manually calculated in a z-stack and the distribution pattern was represented as the percentage of basally or apically located ptf1a+ cells to the total number of ptf1a+ cells. Numbers of strong Pax6+ and Prox1+ cells were calculated using the analyze particles tool in ImageJ and the distribution of cells (INL+ or GCL+) was assigned based on their location relative to the IPL, according to the nuclear staining pattern. Distributional data were represented as the percentage of INL+ or GCL+ to the total number of Prox1+ or Pax6+ cells.

Statistical analysis was conducted using Graphpad Prism 9.1.0. Data are represented as means ± SD. Comparisons between two samples were done using the Mann–Whitney *U*-test or Student's *t*-test with Welch's correction for normally distributed data. For multiple comparisons, two-way analysis of variance with the Tukey post hoc test was used. Details of statistical tests and number of samples used are in figures and figure legends. Significance level is indicated as \*p < 0.05, \*\*p < 0.01, \*\*\*p < 0.001, \*\*\*\*p < 0.0001, n.s. indicates not significant.

## Data availability

Raw RNA-seq data files have been deposited in DDBJ under accession number DRA012640.

MS raw data and result files have been deposited in the ProteomeXchange Consortium (http://proteomecentral.proteomexchange.org) via the jPOST partner repository (https://jpostdb.org) (*Okuda et al., 2017*) under accession number PXD028131.

## Acknowledgements

We are grateful to Rachel Wong for providing the pZNYX-Gal4VP16 construct and *Tg[UAS:MYFP]* fishline and Francesco Argenton for the pG1[ptf1a:GFP] construct. We thank lab members for supporting experiments, especially Yuki Takeuchi, Tetsuya Harakuni, and Yuko Nishiwaki. We thank previous lab members, Fumiyasu Imai, Asuka Yoshizawa, and Sachihiro Suzuki for supporting cloning experiments and fishline generation. We are grateful to Alejandro Villar Briones, formerly of the Instrumental analysis section of the Research Support Division of OIST for supporting mass spectrometry, and the sequencing section of the Research Support Division of OIST for RNA sequencing. This work was funded by a grant from the Okinawa Institute of Science and Technology Graduate University to IM.

## Additional information

### Funding

| Funder | Grant reference number | Author |
| --- | --- | --- |
| Okinawa Institute of Science and Technology Graduate University | | Ichiro Masai |

| Funder | Grant reference number | Author |
|--------|------------------------|--------|

The funders had no role in study design, data collection, and interpretation, or the decision to submit the work for publication.

## Author contributions

Mai Ahmed, Conceptualization, Data curation, Formal analysis, Investigation, Methodology, Project administration, Resources, Software, Validation, Visualization, Writing - original draft, Writing - review and editing; Yutaka Kojima, Investigation; Ichiro Masai, Conceptualization, Data curation, Funding acquisition, Investigation, Methodology, Project administration, Resources, Supervision, Visualization, Writing - review and editing

## Author ORCIDs

Mai Ahmed  http://orcid.org/0000-0002-2952-0646
Ichiro Masai  http://orcid.org/0000-0002-6626-6595

## Ethics

Zebrafish (Danio rerio) were maintained on a 14:10 hr light:dark cycle at 28°C. Collected embryos were cultured in E3 embryo medium (5 mM NaCl, 0.17 mM KCl, 0.33 mM $CaCl_2$, 0.33 mM $MgSO_4$) containing 0.003% 1-phenyl-2-thiouera (PTU) to prevent pigmentation and 0.01% methylene blue to prevent fungal growth. All experiments were performed on zebrafish embryos between 36 hpf and 4 dpf prior to sexual differentiation. Therefore, sexes of the embryos could not be determined. All zebrafish experiments were performed in accordance with the Animal Care and Use Program of Okinawa Institute of Science and Technology Graduate School (OIST), Japan, which is based on the Guide for the Care and Use of Laboratory Animals by the National Research Council of the National Academies. The OIST animal care facility has been accredited by the Association for Assessment and Accreditation of Laboratory Animal Care (AAALAC International). All experimental protocols were approved by the OIST Institutional Animal Care and Use Committee.

## Decision letter and Author response

Decision letter https://doi.org/10.7554/eLife.74650.sa1
Author response https://doi.org/10.7554/eLife.74650.sa2

# Additional files

## Supplementary files

• Transparent reporting form

## Data availability

Raw RNA-seq data files have been deposited in DDBJ under accession number DRA012640. Mass spectrometry raw data and result files have been deposited in the ProteomeXchange Consortium (http://proteomecentral.proteomexchange.org) via the jPOST partner repository (https://jpostdb.org) (Okuda et al., 2017) under accession number PXD028131.

The following datasets were generated:

| Author(s) | Year | Dataset title | Dataset URL | Database and Identifier |
|-----------|------|---------------|-------------|-------------------------|
| Ahmed M, Masai I | 2022 | Transcriptome analysis of strip1 mutant and wildtype zebrafish eyes | https://ddbj.nig.ac.jp/resource/sra-submission/DRA012640 | DDBJ, DRA012640 |
| Ahmed M, Masai I | 2022 | Identification of Strip1-interacting partners in zebrafish head proteome | http://proteomecentral.proteomexchange.org/cgi/GetDataset?ID=PXD028131 | ProteomeXchange, PXD028131 |

The following previously published dataset was used:

| Author(s) | Year | Dataset title | Dataset URL | Database and Identifier |
|---|---|---|---|---|
| Xu B, Tang X, Zhang H, Du L, He J | 2020 | Unifying Developmental Programs for Embryonic and Post-Embryonic Neurogenesis in the Zebrafish Retina | https://www.ncbi.nlm.nih.gov/geo/query/acc.cgi?acc=GSE122680 | NCBI Gene Expression Omnibus, GSE122680 |

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

# Appendix 1

### Appendix 1—key resources table

| Reagent type (species) or resource | Designation | Source or reference | Identifiers | Additional information |
|---|---|---|---|---|
| Genetic reagent (*Danio rerio*) | *Tg[hsp:gal4]$^{kca4}$* | PMID: 11850174 | ZDB-ALT-020918-6 | Reugels/Campos-Ortega lab (Köln University) |
| Genetic reagent (*Danio rerio*) | *Tg[UAS:EGFP]* | PMID: 11336499 | N/A | |
| Genetic reagent (*Danio rerio*) | *Tg[UAS:MYFP]* | PMID: 1702063 | N/A | |
| Genetic reagent (*Danio rerio*) | *Tg[ath5:GFP]$^{rw021}$* | PMID: 12702661 | N/A | |
| Genetic reagent (*Danio rerio*) | *Tg[ptf1a:mCherry-CAAX]$^{oki067}$* | This paper | N/A | See 'Materials and methods' |
| Genetic reagent (*Danio rerio*) | *Tg[Gal4-VP16,UAS:EGFP]xfz43* or *xfz43* | PMID: 19712466 | ZDB-ALT-100201-1 | ZIRC |
| Genetic reagent (*Danio rerio*) | *Tg[Gal4-VP16,UAS:EGFP]xfz3* or *xfz3* | PMID: 19712466 | ZDB-ALT-100201-2 | ZIRC |
| Genetic reagent (*Danio rerio*) | *Tg[hs:mCherry-tagged Bcl2]$^{oki029}$* | PMID: 33060680 | ZDB-ALT-210524-5 | |
| Genetic reagent (*Danio rerio*) | *Tg[hsp:WT.Strip1-GFP]$^{oki068}$* | This paper | N/A | See 'Materials and methods' |
| Genetic reagent (*Danio rerio*) | *Tg[hsp:Mut.Strip1-GFP]$^{oki069}$* | This paper | N/A | See 'Materials and methods' |
| Genetic reagent (*Danio rerio*) | *strip1$^{rw147}$* | This paper | N/A | See 'Materials and methods' |
| Genetic reagent (*Danio rerio*) | *strip1$^{crisprΔ10}$* or *strip1$^{oki8}$* | This paper | N/A | See 'Materials and methods' |
| Genetic reagent (*Danio rerio*) | *roy* | PMID: 28760346 | ZDB-GENE-040426-1168 | |
| Antibody | anti-acetylated α-tubulin (mouse monoclonal) | Sigma-Aldrich | T6793 | IF: 1:1000 |
| Antibody | anti-Pax6 (rabbit polyclonal) | BioLegned | 901,301 | IF: 1:500 |
| Antibody | anti-Prox1 (rabbit polyclonal) | Genetex | GTX128354 | IF: 1:500 |
| Antibody | anti-PCNA (mouse monoclonal) | Sigma-Aldrich | P8825 | IF: 1:200 |
| Antibody | Zpr1 (mouse monoclonal) | ZIRC | ZDB-ATB-081002-43 | IF: 1:100 |
| Antibody | Zpr3 (mouse monoclonal) | ZIRC | ZDB-ATB-081002-45 | IF: 1:100 |
| Antibody | anti-glutamine synthetase (mouse monoclonal) | Sigma-Aldrich | MAB302 | IF: 1:150 |
| Antibody | Zn5 (mouse monoclonal) | ZIRC | ZDB-ATB-081002-19 | IF: 1:50 |
| Antibody | anti-parvalbumin (mouse monoclonal) | Merck Millipore | MAB1572 | IF: 1:500 |
| Antibody | anti-p-Jun (rabbit polyclonal) | Cell Signaling | 9164S | IF: 1:100 |
| Antibody | anti-Strip1 (rabbit polyclonal) | This paper | N/A | See 'Materials and methods' IF: 1:1000 WB: 1:500 |

*Appendix 1 Continued on next page*

*Appendix 1 Continued*

| Reagent type (species) or resource | Designation | Source or reference | Identifiers | Additional information |
|---|---|---|---|---|
| Antibody | anti-rabbit Alexa 488 secondary antibody (goat polyclonal) | Life Technologies | A11034 | IF: 1:500 |
| Antibody | anti-mouse Alexa 488 secondary antibody (goat polyclonal) | Life Technologies | A11029 | IF: 1:500 |
| Antibody | anti-mouse Alexa 546 secondary antibody (goat polyclonal) | Life Technologies | A11030 | IF: 1:500 |
| Antibody | anti-mouse Alexa 647 secondary antibody (goat polyclonal) | Life Technologies | A21236 | IF: 1:500 |
| Antibody | anti-rabbit IgG, HRP-linked Antibody (goat polyclonal) | Cell Signaling | 7074 | WB: 1:5000 |
| Antibody | anti-GFP (rabbit polyclonal) | Thermo Fisher Scientific | A11122 | WB: 1:500 |
| Antibody | anti-Strn3 (rabbit polyclonal) | Thermo Fisher Scientific | PA5-31368 | WB: 1:1000 |
| Antibody | anti β-actin (mouse monoclonal) | Merck Millipore | A5441 | WB: 1:5000 |
| Antibody | anti β-actin (rabbit polyclonal) | Abcam | AB8227 | WB: 1:5000 |
| Recombinant DNA reagent | pTol2[ptf1a:mCherry-CAAX] (plasmid) | This paper | N/A | See 'Materials and methods' |
| Recombinant DNA reagent | pG1[ptf1a:GFP] (plasmid) | PMID: 23035102 | N/A | Francesco Argenton Laboratory (University of Padova) |
| Recombinant DNA reagent | pTol2[hsp:WT.Strip1-GFP] (plasmid) | This paper | N/A | See 'Materials and methods' |
| Recombinant DNA reagent | pTol2[hsp:Mut.Strip1-GFP] (plasmid) | This paper | N/A | See 'Materials and methods' |
| Recombinant DNA reagent | pBluescript SK (+) (plasmid) | Stratagene | N/A | |
| Recombinant DNA reagent | pT2AL200R150G (plasmid) | PMID: 16959904 | N/A | Dr. Koichi Kawakami (Institute of Genetics) |
| Recombinant DNA reagent | pB[ath5:Gal4-VP16] (plasmid) | This paper | N/A | See 'Materials and methods' |
| Recombinant DNA reagent | pZNYX-Gal4VP16 (plasmid) | PMID: 17020638 | N/A | Rachel Wong Laboratory (University of Washington) |
| Sequence-based reagent | Standard control MO (STD-MO) | GeneTools | N/A | 5'-CCTCTTACCTCAGTTACAATTTATA-3' Same concentration for each MO experiment |
| Sequence-based reagent | Strip1 morpholino (MO-strip1) | GeneTools | N/A | 5'- TAGCACAT AAACCGACACCGTCCAT-3' 250 µM |

*Appendix 1 Continued on next page*

*Appendix 1 Continued*

| Reagent type (species) or resource | Designation | Source or reference | Identifiers | Additional information |
|---|---|---|---|---|
| Sequence-based reagent | Ath5 morpholino (MO-ath5) | GeneTools | ZDB-MRPHLNO-100405-2 | 5'-TTCATGGC TCTTCAAAAAAG TCTCC-3' 250 µM |
| Sequence-based reagent | Striatin3 morpholino (MO-strn3) | GeneTools | N/A | 5'- CCTGCTAG AAGTCGCCGATT GTTAC -3' 250 µM |
| Sequence-based reagent | Jun morpholino (MO-jun) | GeneTools | ZDB-MRPHLNO-080908-1 | 5'- CTTGGTAG ACATAGAAGGCA AAGCG -3' 125 µM |
| Peptide, recombinant protein | Cas9 protein | FASMAC | GE-006-S | |
| Commercial assay or kit | JB-4 Embedding Kit | Polysciences | 00226-1 | |
| Commercial assay or kit | In Situ Cell Death Detection Kit, TMR Red | Roche | 12156792910 | |
| Commercial assay or kit | In Situ Cell Death Detection Kit, Fluorescein | Roche | 11684795910 | |
| Commercial assay or kit | DIG RNA Labeling Kit | Roche | 11277073910 | |
| Commercial assay or kit | GFP Trap Agarose | Chromotek | gta-20 | |
| Commercial assay or kit | Arcturus PicoPure RNA Isolation Kit | Thermo Fisher Scientific | KIT0204 | |
| Commercial assay or kit | NEB Next Ultra II Directional RNA Library Prep Kit | New England BioLabs | E7760L | |
| Chemical compound, drug | Acridine Orange (AO) | Nacalai tesque | 1B-307 | 5 µg/ml |
| Chemical compound, drug | CellTrace BODIPY TR Methyl Ester | Thermo Fisher Scientific | C34556 | 100 nM |
| Chemical compound, drug | Ethyl-3-aminobenzoate de methanesulfonate (Tricaine, MS-222) | Nacalai tesque | 14805-82 | 0.02% |
| Chemical compound, drug | PTU (N-Phenylthiourea) | Nacalai tesque | 27429-22 | 0.003% |
| Chemical compound, drug | Fast DiO solid | Thermo Fisher Scientific | D3898 | 2 mg/ml |
| Chemical compound, drug | Fast DiI solid | Thermo Fisher Scientific | D7756 | 2 mg/ml |
| Chemical compound, drug | TO-PRO-3 Iodide (642/661) | Thermo Fisher Scientific | T3605 | 1 nM |
| Chemical compound, drug | Hoechst 33,342 | Wako | 346-07951 | 1 ng/ml |
| Chemical compound, drug | Toluidine Blue | Nacalai tesque | 1B-481 | |
| Chemical compound, drug | Dextran, Tetramethylrhodamine | Thermo Fisher Scientific | D1817 LTJ | |

*Appendix 1 Continued on next page*

*Appendix 1 Continued*

| Reagent type (species) or resource | Designation | Source or reference | Identifiers | Additional information |
|---|---|---|---|---|
| Chemical compound, drug | Dextran, Alexa Flour-488 | Thermo Fisher Scientific | D22910 | |
| Chemical compound, drug | Dextran, Alexa Flour-647 | Thermo Fisher Scientific | D22914 | |
| Chemical compound, drug | Dextran, Cascade Blue | Thermo Fisher Scientific | D1976 | |
| Software, algorithm | chopchop | chopchop | https://chopchop.cbu.uib.no | |
| Software, algorithm | ImageJ (Fiji) | PMID: 22930834 | https://imagej.nih.gov/ij/; RRID: SCR_003070 | |
| Software, algorithm | Imaris | Bitplane | http://www.bitplane.com/imaris; RRID: SCR_007370 | |
| Software, algorithm | Proteome Discoverer | Thermo | https://www.thermofisher.com/store/products/OPTON-30945#/OPTON-30945 | |
| Software, algorithm | STRING | PMID: 27924014 | https://string-db.org | |
| Software, algorithm | Metascape | PMID: 30944313 | https://metascape.org | |
| Software, algorithm | BioVenn | PMID: 18925949 | http://www.biovenn.nl/ | |
| Software, algorithm | GraphPad Prism v9.1.0. | GraphPad Prism | https://www.graphpad.com/scientific-software/prism/ | |

