## [Editor Report]

The results provide mechanistic insight into Strip1 and Striatin-interacting phosphatase and kinase (STRIPAK) complex function at the cellular and molecular level in the developing retina. They show that a primary function of Strip1 and the larger STRIPAK complex in retinal ganglion cells is to promote survival by suppressing Jun-mediated apoptosis. Reviewers were most interested to know whether Jun-mediated, pro-apoptotic signaling occurs due to connectivity defects or if it is connectivity-independent, and the authors have recognized the difficulty in addressing this point, and conclude that it is unlikely that failure of connectivity in the inner plexiform layer is the cause of retinal ganglion cell death.

---

## [Decision Letter]

**Decision letter after peer review:**

Thank you for submitting your work entitled "Strip1 regulates retinal ganglion cell survival by suppressing Jun-mediated apoptosis to promote retinal neural circuit formation" for further consideration by *eLife*. Your article has been evaluated by Marianne Bronner (Senior Editor) and Joseph Gleeson as the Reviewing Editor and three reviewers. The following individual involved in review of your submission has agreed to reveal their identity: Katie Kindt (Reviewer #3).

The reviewers were unanimous in their assessment that there is a substantial amount of novel insight in the manuscript, but remain unconvinced on one major point: Does Strip1 and the larger STRIPAK directly or indirectly regulate signaling and apoptosis. The rationale is that RGCs may require synaptic connection and retrograde signaling to promote survival. The reviewers request that you consider this question in your revision, and provide evidence to better support your model, or at least recognize that the connection could be indirect rather than direct in the text of the manuscript.

*Reviewer #1:*

The authors' presentation of their data in the figures and text is largely clear, and most of the experimental approaches and data analyses are rigorous. The results provide mechanistic insight into Strip1 and STRIPAK function at the cellular and molecular level that will likely be of interest to a wide audience. Because the role of Strip1 in neurite development and the importance of RGCs in retinal lamination have been reported previously, the main novelty of this work lies in the identification of a pathway promoting RGC survival through suppression of Jun signaling. However, the authors' model is weakly supported by their data, and they have not effectively ruled out alternative interpretations of their results. To strengthen their conclusions, these concerns need to be addressed either through clarifications in the text or with additional experiments.

1) The authors' model proposes that a primary function of Strip1 and the larger STRIPAK complex in RGCs is to promote survival by suppressing Jun-mediated apoptosis. However the evidence for this appears largely circumstantial: in strip1 mutants jun mRNA expression is increased, p-Jun levels are elevated, and jun knockdown rescues RGC loss. None of these results demonstrate direct regulation of Jun expression or activity by Strip1, and they arise during the period when IPL defects are first observed and RGCs disappear. It thus seems equally if not more likely that defects in RGC connectivity could be the trigger for Jun activation and apoptosis. Such a mechanism would only require the known role for Strip1 in neurite development, without the need to invoke a separate function acting directly on Jun. The authors should either rule out such a mechanism experimentally, or explain why it was excluded from their model.

2) While the transplant experiments support a cell autonomous role for Strip1 in RGC survival, and a non-cell autonomous role in AC/BP neurite patterning, the cell autonomy of dendrite patterning in RGCs is not addressed. Based on comment #1, it seems important to make this determination, due to the possibility that cell RGC neurite patterning defects could affect connectivity and trigger apoptosis.

3) The section titled "Strip1 interacts with Striatin3 to regulate RGC survival" does not include direct evidence supporting that statement, or the conclusion that the 2 proteins act cooperatively. Strip1 interacts with Striatin3, and both molecules appear to be important for RGC survival, but I don't see any experiments specifically testing whether the interaction itself is required for RGC survival, or testing epistasis between the two genes. This wording should either be changed to accurately represent the data, or supported with additional experiments.

There are errors in English spelling, word usage, and grammar throughout the manuscript, which would benefit from editing by a native speaker.

*Reviewer #2:*

Ahmed et al., study the role of Strip1 in the development of the inner retina, specifically the survival of the RGCs and the establishment of the inner plexiform layer (IPL). The authors show a cell intrinsic role for Strip1 in the survival of RGCs soon after cell birth. Strip1 is part of the STRIPAK complex and the authors show it functions through interaction with striatin3 causing a repression of the pro-apoptotic transcription factor c-Jun. Although Strip1 is also expressed in amacrine cells the author's show that its pro-survival role is limited to the RGCs. Loss of the RGCs leads to a breakdown in the patterning structure of the inner retina leading to a loss of the IPL and mislocalization of both amacrine cells and bipolar cells. Amacrines and bipolars also show abnormal neurite projections in Strip1 knockouts.

The use of blastula transplants of mutant and wildtype cells is a very strong and elegant experimental design to show cell intrinsic functions of Strip1 in RGCs. These experiments along with their characterization of the Strip1 mutants provide solid evidence for the claims that Strip1 is a key component in cell survival of RGC during development.

Major corrections:

The authors claim that data from figure 7 "suggest an additional role of Strip1 in dendritic patterning of RGCs, which is likely to prevent ectopic IPL-like neuropil formation in the AC layer." And subsequently discuss potential developmental models for the interaction of ACs and RGCs to establish the IPL.

The current data is insufficient to make these claims and further experiments could clarify this issue. In figure 7C the lack of RGC dendritic projections observed in the strip1rw147 Bcl2 rescue could also be due to the loss of strip1 function in the ACs. One potential way to directly test if the role of strip1 in RGC dendrite formation is a RGC intrinsic mechanism is to analyze the dendritic projections of the surviving RGCs in the mutant to wildtype blastula transplants in figure 3E. If it were a cell intrinsic mechanism you would see lack of dendrite formation in the surviving mutant RGCs in the wildtype host.

*Reviewer #3:*

In this study, Ahmed, Kojima and Masai, investigate how the retinal inner plexiform layer (IPL) is formed during development. The IPL houses the synaptic connections between retinal ganglion cells (RGCs), amacrine cells (ACs) and bipolar cells (BCs), which are required to form neural circuits critical for vision. The authors first identify the involvement of strip1 in IPL formation through a forward-genetic screen, and confirm this finding with 2 independent approaches: creation of a strip1 mutant using CRISPR-guided mutagenesis and strip1 translation block using morpholinos. They show that strip1 is expressed by RGCs and ACs, and that, perhaps unexpectedly, the IPL defects in strip1 mutants stem primarily from RGC death. By tracking defects throughout development and generating chimeric animals, the authors establish that other retinal cells, including ACs, are unaffected by strip1 loss-of-function, that Strip1 is required for RGC survival cell-autonomously (but not for RGC genesis), and that the invasion of the IPL and ganglion-cell layer by ACs is secondary to the RGC loss. In addition, the authors identify Strn3 as a Strip1 binding partner with overlapping function in RGC survival, and Jun as a downstream mediator of RGC death in strip1 mutants. Finally, the authors inhibit RGC death in strip1 mutants and discover that IPL formation is still abnormal, suggesting a second role of Strip1 in the correct formation of RGC dendrites.

There are many strengths to this manuscript. It is written in a clear and concise manner. The figures are informative and the images are beautiful. The findings are well supported, with appropriate controls. Each result is confirmed with several independent techniques when possible. The interpretations are presented in significant contexts. In addition, the authors push experiments and conclusions further by making use of technical advantages that are specific to the zebrafish model including: forward-genetic screening, tracking retinal development in a model that is embryonic-lethal in mice, using transgenic lines for in vivo analyses or targeted overexpression and rescue and, finally, creating genetic chimeras to explore autonomy of Strip1.

There are also several weaknesses or points that readers should consider while reading this work. In a secondary experiment the authors examine the role of Strn3, a potential Strip1 interactor. The morpholino-manipulation of strn3 lacks the controls used previously for strip1 (use of an antibody to confirm specificity of knockdown and germline strn3 mutants). Many morpholinos are known to cause apoptosis/cell-death irrespective of the gene they intend to target. The reader should consider this potential caveat. The authors claim Strip1 interacts with Strn3 to suppress RGC cell death. But the IPL defects are not well correlated with degree of RGC loss between strip1 mutants and strn3 morphants. Therefore, it possible that the specific subtypes of RGC that are lost with these manipulations are different. Lastly, the reader may consider the possibility that the RGC degeneration is accompanied by regeneration. Similar to strip1 mutants, a specific rhodopsin mutation causes death of rod photoreceptors without specification defects (https://www.ncbi.nlm.nih.gov/pmc/articles/PMC7599532/). In this study, some of the late structural defects are thought to arise from continuous waves of degeneration and regeneration. Therefore, it is possible that some of the defects in strip1 mutants may also be due to an increase in regeneration.

Overall the data presented in this manuscript is likely to be of direct importance for two main fields-retinal disease and development-and has the potential to generate further questions and studies. In the context of retinal disease, as the authors point out, secondary loss of RGCs is an important cause of blindness. Understanding manipulations that promote RGC survival is critical for therapeutical applications. The discovery of Strip1 and other components of the STRIPAK complex as regulators of RGC survival is important. Regarding retinal development, it is no surprise that the formation of the IPL is a complex process, as a myriad of amacrine, bipolar and ganglion cells must wire with their specific synaptic patterns to form multiple neural circuits to extract visual information. Although RGCs are born first, it is thought that IPL formation relies first on ACs extending processes that serve as anchors for other cells. In this study, Ahmed, Kojima and Masai reveal that this initial process is more complex, and requires active coordination between ACs and RGCs, beyond RGCs occupying space to prevent misplacement of ACs. In the future, it will be interesting to determine if particular subtypes of ACs and RGCs are especially important for IPL formation, while others "follow along".

The data presented in the work is comprehensive and well-controlled. We do not recommend any further experiments or changes to the presentation.

---

## [Author Response]

Reviewer #1:The authors' presentation of their data in the figures and text is largely clear, and most of the experimental approaches and data analyses are rigorous. The results provide mechanistic insight into Strip1 and STRIPAK function at the cellular and molecular level that will likely be of interest to a wide audience. Because the role of Strip1 in neurite development and the importance of RGCs in retinal lamination have been reported previously, the main novelty of this work lies in the identification of a pathway promoting RGC survival through suppression of Jun signaling. However, the authors' model is weakly supported by their data, and they have not effectively ruled out alternative interpretations of their results. To strengthen their conclusions, these concerns need to be addressed either through clarifications in the text or with additional experiments.1) The authors' model proposes that a primary function of Strip1 and the larger STRIPAK complex in RGCs is to promote survival by suppressing Jun-mediated apoptosis. However the evidence for this appears largely circumstantial: in strip1 mutants jun mRNA expression is increased, p-Jun levels are elevated, and jun knockdown rescues RGC loss. None of these results demonstrate direct regulation of Jun expression or activity by Strip1, and they arise during the period when IPL defects are first observed and RGCs disappear. It thus seems equally if not more likely that defects in RGC connectivity could be the trigger for Jun activation and apoptosis. Such a mechanism would only require the known role for Strip1 in neurite development, without the need to invoke a separate function acting directly on Jun. The authors should either rule out such a mechanism experimentally, or explain why it was excluded from their model.

We thank reviewer #1 for drawing our attention to this important point. It is difficult to determine whether Jun-mediated, pro-apoptotic signaling occurs due to connectivity defects or if it is connectivity-independent. Indeed, we found that Strip1 is important for RGC neurite development (please see further details in our response to comment #2). However, our data at the cellular and molecular levels suggest that the apoptotic program may commence prior to initiation of normal connectivity in the IPL. We have added additional data that shows activation of Jun in strip1 mutant RGCs at 48 hpf (Figure 6—figure supplement 2), at the same timepoint we begin to observe apoptosis in the GCL using TUNEL (Figure 3). On the other hand, previous reports suggest that RGCs start to project apical dendrites and innervate the IPL at later stages, around 55-60 hpf, following lamination cues from ACs (Choi et al., 2010, Mumm et al., 2006). In addition, synaptogenesis in the IPL starts at around 60 hpf (Schmitt and Dowling, 1999). Furthermore, our time-lapse imaging shows that RGC death starts prior to IPL malformation (Figure 4A). Therefore, it is unlikely that failure of connectivity in the IPL is the cause of RGC death.

On the other hand, understanding the contribution of possible connectivity defects in the optic tectum to RGC death is more challenging. In wild-type zebrafish embryos, complete optic nerve transection in 5-dpf larvae does not induce substantial RGC death (Harvey et al., 2019). However, at 48 hpf, we observe that Jun activation starts in the earliest-born retinal neurons, which coincides with the timing when wildtype early-born RGCs start innervating the optic tectum (Burrill and Easter Jr, 1994, Stuermer, 1988). It is possible that in wildtype zebrafish embryos, when connectivity to the optic tectum is compromised, functional Strip1-mediated survival machinery prevents stress-induced RGC apoptosis. However, in strip1 mutants, this survival machinery is disrupted, leading to RGC death. We have revised the Discussion section accordingly to cover such possibilities (Line 471-497).

2) While the transplant experiments support a cell autonomous role for Strip1 in RGC survival, and a non-cell autonomous role in AC/BP neurite patterning, the cell autonomy of dendrite patterning in RGCs is not addressed. Based on comment #1, it seems important to make this determination, due to the possibility that cell RGC neurite patterning defects could affect connectivity and trigger apoptosis.

We thank reviewer #1 for this suggestion. We added additional experiments to address the cell autonomy of Strip1 in RGC dendritic patterning. We performed cell transplantation experiments using donor strip1 mutants carrying the transgene ath5:GFP to label RGCs, and transplanted them into wild-type hosts. We observed that most transplanted mutant RGCs display neurite projection defects (multiple distant dendritic branches rather than a normal, uniform pattern of apically projected dendrites). Therefore, we concluded that Strip1 is cell-autonomously required for RGC neurite projections, and we included this data in a new figure (Figure 3—figure supplement 2). As we mentioned in response to comment #1, understanding the impact of such connectivity defects on the observed apoptosis is challenging because of the early and progressive degeneration of RGCs. Therefore, it is possible that both connectivity-dependent and -independent mechanisms trigger pro-apoptotic signaling in RGCs of strip1 mutants. As we mentioned above, we have revised the Discussion section accordingly to cover such possibilities (Line 471-497).

3) The section titled "Strip1 interacts with Striatin3 to regulate RGC survival" does not include direct evidence supporting that statement, or the conclusion that the 2 proteins act cooperatively. Strip1 interacts with Striatin3, and both molecules appear to be important for RGC survival, but I don't see any experiments specifically testing whether the interaction itself is required for RGC survival, or testing epistasis between the two genes. This wording should either be changed to accurately represent the data, or supported with additional experiments.

We agree with reviewer #1 that this title somewhat overstated the findings. We have revised the description and our conclusion for Striatin 3 data in the Abstract, Results, Discussion, and figure legends to accurately represent the findings that Strip1 interacts with Striatin3, and they both show overlapping roles in RGC survival.

There are errors in English spelling, word usage, and grammar throughout the manuscript, which would benefit from editing by a native speaker.

The manuscript has been proofread by a professional technical editor who is a native speaker. We hope that any errors have been fully corrected.

Reviewer #2:Major corrections:The authors claim that data from figure 7 "suggest an additional role of Strip1 in dendritic patterning of RGCs, which is likely to prevent ectopic IPL-like neuropil formation in the AC layer." And subsequently discuss potential developmental models for the interaction of ACs and RGCs to establish the IPL.The current data is insufficient to make these claims and further experiments could clarify this issue. In figure 7C the lack of RGC dendritic projections observed in the strip1rw147 Bcl2 rescue could also be due to the loss of strip1 function in the ACs. One potential way to directly test if the role of strip1 in RGC dendrite formation is a RGC intrinsic mechanism is to analyze the dendritic projections of the surviving RGCs in the mutant to wildtype blastula transplants in figure 3E. If it were a cell intrinsic mechanism you would see lack of dendrite formation in the surviving mutant RGCs in the wildtype host.

We thank reviewer #2 for this positive feedback and we agree that the Bcl2 rescue data aren’t sufficient to draw conclusions about the importance of Strip1 in dendritic patterning of RGCs. As we mentioned in comment #2 for reviewer #1, we have performed additional experiments to address the cell autonomy of Strip1 in RGC dendritic patterning. In accordance with the referee’s suggestion, we repeated cell transplantation experiments using donor strip1 mutants carrying the transgene ath5:GFP to label RGCs and transplanted them into wild-type hosts. We observe that most transplanted mutant RGCs display neurite projection defects (multiple distant dendritic branches rather than the normal uniform pattern of apically projected dendrites). Therefore, we concluded that Strip1 is cell autonomously required for proper RGC neurite projections, and we included this data in a new figure (Figure 3figure supplement 2) and revised the manuscript accordingly.

Reviewer #3:In this study, Ahmed, Kojima and Masai, investigate how the retinal inner plexiform layer (IPL) is formed during development. The IPL houses the synaptic connections between retinal ganglion cells (RGCs), amacrine cells (ACs) and bipolar cells (BCs), which are required to form neural circuits critical for vision. The authors first identify the involvement of strip1 in IPL formation through a forward-genetic screen, and confirm this finding with 2 independent approaches: creation of a strip1 mutant using CRISPR-guided mutagenesis and strip1 translation block using morpholinos. They show that strip1 is expressed by RGCs and ACs, and that, perhaps unexpectedly, the IPL defects in strip1 mutants stem primarily from RGC death. By tracking defects throughout development and generating chimeric animals, the authors establish that other retinal cells, including ACs, are unaffected by strip1 loss-of-function, that Strip1 is required for RGC survival cell-autonomously (but not for RGC genesis), and that the invasion of the IPL and ganglion-cell layer by ACs is secondary to the RGC loss. In addition, the authors identify Strn3 as a Strip1 binding partner with overlapping function in RGC survival, and Jun as a downstream mediator of RGC death in strip1 mutants. Finally, the authors inhibit RGC death in strip1 mutants and discover that IPL formation is still abnormal, suggesting a second role of Strip1 in the correct formation of RGC dendrites.There are many strengths to this manuscript. It is written in a clear and concise manner. The figures are informative and the images are beautiful. The findings are well supported, with appropriate controls. Each result is confirmed with several independent techniques when possible. The interpretations are presented in significant contexts. In addition, the authors push experiments and conclusions further by making use of technical advantages that are specific to the zebrafish model including: forward-genetic screening, tracking retinal development in a model that is embryonic-lethal in mice, using transgenic lines for in vivo analyses or targeted overexpression and rescue and, finally, creating genetic chimeras to explore autonomy of Strip1.

We thank reviewer #3 for this positive feedback.

There are also several weaknesses or points that readers should consider while reading this work. In a secondary experiment the authors examine the role of Strn3, a potential Strip1 interactor. The morpholino-manipulation of strn3 lacks the controls used previously for strip1 (use of an antibody to confirm specificity of knockdown and germline strn3 mutants). Many morpholinos are known to cause apoptosis/cell-death irrespective of the gene they intend to target. The reader should consider this potential caveat.

We thank reviewer #3 for raising these concerns. We agree with the reviewer’s comments regarding the weakness of MO-strn3 data due to lack of validation data. We have further strengthened our findings by validating this morpholino using a commercial antibody to confirm the reduction of a Strn3-specific band in 2-dpf morphants. We included this data in a new figure (Figure 5—figure supplement 2).

The authors claim Strip1 interacts with Strn3 to suppress RGC cell death. But the IPL defects are not well correlated with degree of RGC loss between strip1 mutants and strn3 morphants. Therefore, it possible that the specific subtypes of RGC that are lost with these manipulations are different.

We thank the reviewer for drawing our attention to this point. We were surprised that strn3 morphants did not display IPL defects comparable to those of strip1 mutants. Although MO-strn3 reduces the RGC population significantly, this reduction is also weaker than that of strip1 mutants. Therefore, the surviving RGC population in strn3 morphants at 3 dpf might be sufficient to maintain the structural integrity of the IPL. As the reviewer suggested, it is possible that Strip1 and Strn3 affect different RGC subtypes. Since we investigate IPL defects at 3 dpf, another possibility is that the morpholino concentration was slightly diluted by this stage, leading to a weaker phenotype. We have included such possibilities in the Results section of the revised manuscript.

Lastly, the reader may consider the possibility that the RGC degeneration is accompanied by regeneration. Similar to strip1 mutants, a specific rhodopsin mutation causes death of rod photoreceptors without specification defects (https://www.ncbi.nlm.nih.gov/pmc/articles/PMC7599532/). In this study, some of the late structural defects are thought to arise from continuous waves of degeneration and regeneration. Therefore, it is possible that some of the defects in strip1 mutants may also be due to an increase in regeneration.

The possible role of Müller glia-mediated regeneration on the observed defects of strip1 mutants is interesting. We haven’t examined this point in our study. However, previous analysis of Müller glia genesis in zebrafish retina suggests that they reach their mature states at around 72 hpf

(https://www.ncbi.nlm.nih.gov/pmc/articles/PMC4586739/). By this timepoint, the layering defects of strip1 mutants are already evident. In addition, PCNA labeling doesn’t show prominent increase of proliferating cells in the INL/GCL of strip1 mutants compared to their wild type siblings, suggesting that it is unlikely that Müller gliamediated regeneration program is activated in strip1 mutants by the stages we examined. Moreover, we observe a steady and progressive degeneration of RGCs until 4 dpf, which is consistent with the progression of cell death. Therefore, regeneration of RGCs may not play a major role in the observed defects of strip1 mutants.

Overall the data presented in this manuscript is likely to be of direct importance for two main fields-retinal disease and development-and has the potential to generate further questions and studies. In the context of retinal disease, as the authors point out, secondary loss of RGCs is an important cause of blindness. Understanding manipulations that promote RGC survival is critical for therapeutical applications. The discovery of Strip1 and other components of the STRIPAK complex as regulators of RGC survival is important. Regarding retinal development, it is no surprise that the formation of the IPL is a complex process, as a myriad of amacrine, bipolar and ganglion cells must wire with their specific synaptic patterns to form multiple neural circuits to extract visual information. Although RGCs are born first, it is thought that IPL formation relies first on ACs extending processes that serve as anchors for other cells. In this study, Ahmed, Kojima and Masai reveal that this initial process is more complex, and requires active coordination between ACs and RGCs, beyond RGCs occupying space to prevent misplacement of ACs. In the future, it will be interesting to determine if particular subtypes of ACs and RGCs are especially important for IPL formation, while others "follow along".

We thank the reviewer for highlighting the significance and broad impact of our findings.

The data presented in the work is comprehensive and well-controlled. We do not recommend any further experiments or changes to the presentation.